



# REEs associated with carbonatite-alkaline complexes in western Rajasthan, India: exploration targeting at regional-scale

Malcolm Aranha[1], Alok Porwal[1,2], Manikandan Sundaralingam[3], Ignacio González-Álvarez[4,2], Amber Markan[3], Karunakar Rao[3].

[1]Centre of Studies in Resources Engineering (CSRE), Indian Institute of Technology Bombay, Mumbai, 400076, India
[2]Centre for Exploration Targeting, University of Western Australia, Crawley, 6009, Australia
[3]Datacode, Nagpur, 440033, India
[4]Commonwealth Scientific and Industrial Research Organisation (CSIRO), Mineral Resources, Kensington, 6151, Australia

*Correspondence to*: Malcolm Aranha (malcolmaranha@iitb.ac.in)

**Abstract.** A two-stage fuzzy inference system (FIS) is applied to prospectivity modelling and exploration-target delineation for REE deposits associated with carbonatite-alkaline complexes in western part of the state of Rajasthan in India. The design of the FIS and selection of the input predictor map are guided by a generalized conceptual model of carbonatite-alkaline-complexes-related REE mineral systems. In the first stage, three FISs are constructed to map the fertility and favourable geodynamic settings, favourable lithospheric architecture, and favourable shallow crustal (near-surface) architecture,
respectively, for REE deposits in the study area. In the second stage, the outputs of the above FISs are integrated to map the prospectivity of REE deposits in the study area. Stochastic and systemic uncertainties in the output prospectivity maps are estimated to facilitate decision making regarding the selection of exploration targets. The study led to identification of prospective targets in the Kamthai-Sarnu-Dandeli and Mundwara regions, where project-scale detailed ground exploration is recommended. Low-confidence targets were identified in the south of the Siwana ring complex, north and northeast of Sarnu-
Dandeli, south of Barmer, and south of Mundwara. Detailed geochemical sampling and high-resolution magnetic and radiometric surveys are recommended in these areas to increase the level of confidence in the prospectivity of these targets before undertaking project-scale ground exploration. The prospectivity-analysis workflow presented in this paper can be applied to delineation of exploration targets in geodynamically similar regions globally such as Afar province (East Africa), Paraná-Etendeka (South America and Africa), Siberian (Russia), East European Craton-Kola (Eastern Europe), Central Iapetus
(North America, Greenland and the Baltic region), and the Pan-superior province (North America).

**Keywords**

Prospectivity Modelling, Uncertainty Modelling, Rare Earth Elements (REE), Carbonatite-Alkaline Complex, Fuzzy Inference System, Western Rajasthan



## 1 Introduction

The term Rare Earth Elements (REEs) includes (International Union of Pure and Applied Chemistry, IUPAC): yttrium (Y), scandium (Sc), and the lanthanides (lanthanum, La; cerium, Ce; praseodymium, Pr; neodymium, Nd; promethium, Pm; samarium, Sm; europium, Eu; gadolinium, Gd; terbium, Tb; dysprosium, Dy; holmium, Ho; erbium, Er; thulium, Tm; ytterbium, Yb; and lutetium, Lu). Because of their increasing use in environment-friendly high-technology industries, REEs

are widely considered as the resources of the future (e.g., Goodenough et al., 2018; Wall, 2021). Most countries have classified REEs as 'critical minerals and metals' because of their strategic importance and the projected gap between their future demand and supply (Goodenough et al., 2018; Gonzalez-Alvarez et al., 2021 and references therein).

  In spite of significant efforts into developing technology for recovering and recycling REEs from discarded devices (Binnemans et al., 2013), geological resources are likely to remain the primary sources of REEs in the foreseeable future

(Goodenough et al., 2018). Several classification schemes for REE deposits have been proposed by different workers based on geological associations and settings; for example, Chakhmouradian and Wall (2012), Jaireth et al. (2014), Wall (2014), Goodenough et al. (2016), Verplanck and Hitzman (2016), Simandl and Paradis (2018), etc. In general, REE deposits can be broadly classified into those formed by high-temperature (magmatic and hydrothermal) processes and those formed by low-temperature (mechanical and residual concentration) processes (e.g., Wall, 2021). Although the majority of the global

production of REEs comes from low-temperature deposits such as regolith-hosted and heavy-mineral placers (IBM yearbook 2018, 2019), the bulk of geological resources are in high-temperature magmatic deposits, particularly those associated with carbonatites (e.g., Bayan Obo, Inner Mongolia, China; Mount Weld, Western Australia; Maoniuping, South China; Mountain Pass, USA etc.; Gonzalez-Alvarez et al., 2021 and references therein)

  India ranks 6th in terms of production of REEs and 5th in terms of resources (USGS, 2021). All of India's production comes

from monazite-bearing beach sands along the eastern and western coasts (IBM yearbook 2018, 2019). Since India has 29 out of the total 527 globally reported carbonatite occurrences (Woolley and Kjarsgaard, 2008a), there is significant latent potential for carbonatite-related REE deposits in the country. Currently, there is no study available, at least in the public domain, on systematic delineation of prospective REE exploration targets in India.

  Mineral prospectivity modelling is a widely used predictive tool for identifying exploration target areas. Implemented in a GIS

environment, it involves the integration of 'predictor maps' that represent a set of mappable exploration criteria for the targeted deposit type. Typically, conceptual mineral systems models are used to identify exploration criteria (Porwal and Kreuzer, 2010; Porwal and Carranza, 2015). The integration is done through either linear or non-linear mathematical functions (Bonham-Carter, 1994; Porwal, 2006; Porwal and Carranza, 2015). Depending on how the model parameters are estimated, that is, whether based on training data comprising attributes of known deposits or on expert knowledge, these models are

classified as data-driven or knowledge-driven.

  Data-driven approaches require a sizeable sample of known deposits of the targeted deposit type for estimating the model parameters. The main data-driven approaches are Bayesian probabilistic approaches (e.g., Weights of Evidence - Singer and





Kouda, 1999; Nykänen et al., 2008; Porwal et al., 2010; Zhang et al., 2014; Nielsen et al., 2015; Payne et al., 2015; Chudasama et al., 2018; Tao et al., 2019), regression-based approaches (e.g., Logistic regression - Harris and Pan, 1999; Carranza and

Hale, 2001; Harris et al., 2003; Nykänen et al., 2008; Porwal et al., 2010; Chen et al., 2011; Zhang et al., 2014; Xiong and Zuo, 2018) and machine learning approaches (e.g., neural networks – Singer and Kouda, 1999; Brown et al., 2000; Porwal et al., 2003a; Rodriguez-Galiano et al., 2015; Chudasama et al., 2018; Sun et al., 2020; Support Vector Machines - Zuo and Carranza, 2011; Abedi et al., 2012; Rodriguez-Galiano et al., 2015; Chen and Wu, 2017; Random forests - Rodriguez-Galiano et al., 2015; Carranza and Laborte, 2015; Hariharan et al., 2017). Knowledge-driven approaches use expert knowledge for

estimating model parameters. Typical knowledge-driven approaches include fuzzy-set theory-based expert systems (Porwal et al., 2003b; Nykänen et al., 2008; González-Álvarez et al., 2010; Joly et al., 2012; Porwal et al., 2015; Wilde et al., 2018; Chudasama et al., 2018; Morgenstern et al., 2018), Dempster–Shafer evidential belief functions (Moon, 1990, 1993; An et al., 1994; Chung and Fabbri, 1993; Tangestani and Moore, 2002; Carranza and Sadeghi, 2010).

Prospectivity modelling is subject to uncertainties. These uncertainties are classified in two main categories (Porwal et al.,

2003b; Lisitsin et al., 2014), namely, systemic and stochastic. Systemic uncertainties rise from the incomplete understanding of the geological process involved in the formation of the mineral deposit, leading to imperfect or inefficient models. Stochastic uncertainties rise from the limitations of datasets used, of the methods used to interpret useful information from them. These can be a result of inaccurate or imprecision of measurements, mapping or interpolations, inconsistent data coverage etc (Porwal et al., 2003b; McCuaig et al., 2009; Lisitsin et al., 2014).

There are very few published studies on REE prospectivity modelling. Ekmann (2012) escorted a study of REEs in coal deposits in the United States. In one of the first GIS-based prospectivity modelling studies for REEs, Aitken et al. (2014) used a fuzzy-logic-based model to delineate prospective targets for pegmatite-, carbonatite- and vein-hosted REEs in the Gascoyne Region of Western Australia. This study was part of a larger multi-commodity prospectivity study of the Gascoyne Province. Sadeghi (2017) carried out a regional-scale GIS-based prospectivity analysis for REEs in the Bergslagen district of Sweden,

targeting iron-apatite- and skarn-associated deposits using the weights of evidence and weighted-overlay models. Bertrand et al. (2017) used database querying to analyse the prospectivity for REEs as by-products in known mineral deposits in Europe. In a recent study, Morgenstern et al. (2018) analysed the potential of REEs in New Zealand using a multi-stage Fuzzy inference system (FIS).

This contribution describes the first systematic and comprehensive prospectivity modelling exercise aimed at identifying

exploration targets for REE associated with carbonatite-alkaline complexes in Western Rajasthan, India (Fig. 1). Although a well-established carbonatite province that is widely considered prospective for REE deposits, no deposit has been identified in the province so far. In this study, we employ fuzzy inference system (FIS), which is a knowledge-driven artificial intelligence technique, to identify and delineate prospective targets for REE deposits in the study area. The inputs to the FIS were identified based on a generalised mineral systems model for alkaline-carbonite-complexes-related REEs, which was further used to guide

the design of the FIS. To support decision making regarding the delineated targets, stochastic and systemic uncertainties in the output model were also estimated. The prospectivity-analysis workflow presented in this paper can be applied to other





geodynamically similar regions globally for targeting regions for follow-up exploration in East Africa, South and North America, Russia, Eastern Europe, Greenland, and the Baltic region.

## 2 Geological setting of the study area

The study area falls in the state of Rajasthan in northwest India (Fig. 1). This area was chosen because it is a known major carbonatite province of India, and well-integrated public domain datasets are available. Geologically, the study area contains igneous and sedimentary formations ranging in age from the Neoproterozoic to Holocene. Neoproterozoic Erinpura and Jalore granites, along with a few outcrops of the Mesoproterozoic Delhi Supergroup, occur in the southeastern part of the study area (Fig. 1). The eastern part of the study area comprises extrusive and intrusive igneous rocks belonging to the Neoproterozoic

Malani Igneous Suite that is mostly buried under a thick horizon of Holocene wind-blown sand. Sedimentary sequences belonging to the Late Neoproterozoic Marwar Supergroup, Jurassic Jaisalmer, Cretaceous Sarnu-Fatehgarh, Tertiary Barmer (Palaeocene) and Akli (Eocene), Quaternary Uttarlai Formations (Pleistocene to sub-Recent) (Roy and Jakhar, 2002; Ramakrishnan and Vaidyanadhan, 2008; Singh et al., 2016) occur in the central and western parts around Barmer and Jaisalmer towns (Fig. 1).

Carbonatite-alkaline complexes of the Cretaceous age occur in the Mer-Mundwara area in the eastern part of the study area and the Sarnu-Dandali area in the central part (Fig. 1; Table 1). The Mer-Mundwara carbonatite-alkaline complex intrudes the Neoproterozoic Erinpura Granite and displays a characteristic ring structure, wherein the alkaline-mafic rock suites form two ring structures and a dome (Pande et al., 2017). Carbonatites mainly occur in the form of linear dykes at Mer-Mundwara. The Sarnu-Dandeli complex covers a relatively large area on the eastern shoulder of the Barmer basin. The carbonatites occur

mainly as scattered plugs and dykes with an extensive Quaternary sand cover, intruding the Neoproterozoic Malani igneous suite and the Cretaceous Sarnu formation (Vijayan et al., 2016; Sheth et al., 2017). It also includes more minor occurrences of carbonatites in the Danta-Langera-Mahabar and Kamthai areas. The Kamthai plug is considered to be highly prospective for REEs (Bhushan and Kumar, 2013).

The study area is dissected by the Barmer rift, which continues southwards through the state of Gujarat into the Cambay basin.

The Barmer rift is a failed, roughly north-south trending, extensional intracratonic rift (Fig. 1) that was active during Late Cretaceous to Eocene (Dolson et al., 2015). A long-lasting extensional regime in northwest India predating the Deccan volcanism existed in northwest India, peaked with the Seychelles rifting at the Cretaceous–Paleogene boundary and the emplacement of the main phase of Deccan volcanics at ca. 65 Ma (Devey and Stephens, 1992; Allegre et al., 1999; Chenet et al., 2007; Collier et al., 2008; Ganerød et al., 2011; Bladon et al., 2015a, b). The well-preserved Cretaceous carbonatite-alkaline

complexes of the study area represent a young carbonatite magmatism episode (~68 Ma) that is coeval with the initial magmatism of the Deccan Large Igneous Province (LIP) and is related to the India-Seychelles breakup and northward drifting of India over the reunion mantle plume (Devey and Stephens, 1992; Basu et al., 1993; Simonetti et al., 1995; Allegre et al.,



1999; Ray and Pande, 1999; Ray and Ramesh, 1999; Ray et al., 2000; Chenet et al., 2007; Collier et al., 2008; Sheth et al., 2017; Chandra et al., 2018).

**Figure 1: Geological map of the study area with known carbonatite-alkaline complexes.**

## 3 Datasets and methods

The public domain geoscience datasets used in the study, which include geological, geophysical, topographic and satellite data, were sourced from the Bhukosh portal of the Geological Survey of India (https://bhukosh.gsi.gov.in/Bhukosh/MapViewer.aspx). Table 2 summarises the sources, scales and other details about the individual datasets.



**Table 1: Location, physiography and geological setting of alkaline-carbonatite complexes in NW India.**

| Complex / District | Location and physiography | Regional geological setting | Geochemistry | Remarks | Key References |
|---|---|---|---|---|---|
| Sarnu-Dandeli (25.614, 71.884) Danta-Langera-Mahabar (25.73, 71.42; Location and presence uncertain) | • Barmer district, Rajasthan state • located on the eastern shoulder of the Barmer basin. • Carbonatites occur as dykes and veins varying from a few cm to about 12 m in length and from a few mm to about 30 cm in width cutting across the mafic alkaline rocks of the complex • Carbonatite dykes up to 12 m in length and 30 cm thick are found at Danta-Langera-Mahabar (similar to Sarnu Dandeli) • Located in the eastern edge of the Sarnu-Dandali complex in the Barmer basin. • Highly enriched in REE and is considered a potential world-class deposit. | • occurs within the Barmer continental rift basin • intrudes the Neoproterozoic rhyolitic rocks of the Malani Igneous province and Cretaceous sandstones and siltstones that are underlain by a basaltic flow • Barmer rift basin contains thick Mesozoic sediments that contain significant hydrocarbon reserves that are being actively exploited • Barmer and the Cambay basins have formed during Late Cretaceous in response to far-field stresses related to the Gondwana plate reorganizations that led to the formation of the 600 km long intracontinental rift. | • $\delta^{13}C_{‰V-PDB}$ vary between −6.1 and −1.4 • $\delta^{18}O_{‰V-SMOW}$ vary between 28.2 and 8.2 • Some fall within mantle box but most of the calcite carbonatites have higher $\delta^{13}C$ and $\delta^{18}O$ values compared to those of the mantle. • Mantle derived carbonatites based on $\delta^{13}C$ ratios, REE chemistry and the La/Lu ratio. • Average La/Yb ratio 1232.97 • Average values for La, Ce, Pr, and Nd are 1.29%, 1.58%, 0.12% and 0.3% respectively. | | Chandrasekaran, 1987; Chandrasekaran and Chawade, 1990; Chandrasekaran et al., 1990; Ray et al, 2000; Vijayan et al., 2016; Sheth et al., 2017 |
| Kamthai (25.633, 71.931) | • Kamthai plug is ellipsoidal in shape, covering an area of 19,475 sq.m. • Dykes and veins of carbonatites are associated. • Characteristic panther skin texture for the plug carbonatites and golden yellow colour and occasional elephant skin weathering for the dykes and sills has been noted. | | • Average Sm and Eu values are 135 ppm and 43 ppm respectively. | • Total REE resources of 4.65 MT, including 0.66 MT (proven), 1.33 MT (probable) and 2.66 MT (possible), in the Kamthai plug at the average grade of 2.69%. • Additional resources of 259,000 t from dykes, sills and veins. • Overall, a total of 4.91 MT of REE ore at an average grade of 2.97% | Bhushan and Kumar, 2013; Bhushan, 2015 |
| Mer Mundwara (24.828, 72.537) | • Sirohi District, Rajasthan state consists of three laccolith type intrusive plutons, namely, – Musala, Mer and Toa. • Mer is the biggest intrusion consisting of a complete ring structure and is about 1.3 km in diameter. • Toa forms half a ring structure and is about 700 m in diameter. • Musala appears as a mound, about 500 m in diameter. | • Occurs in close vicinity of the Barmer continental rift • Intrudes into the Neoproterozoic Erinpura Granite | • Pressure conditions of magma emplacement and crystallization are estimated to range between 2 and 7 kbar (200–700 MPa) at an assumed mean temperature of 1000°C using a clinopyroxene barometer. • Amphibole thermobarometer indicates pressures of 686–306 MPa (3–7 kbar) and temperatures of 935–1030°C and for kaersutite and pargasite crystallization from melts containing 2.8–6.0 wt% $H_2O$. • Oxygen fugacities (log $fO_2$) vary between −23.9 and −14.2 | | Ray et al., 2000; Pande et al., 2017 |



**Table 2: A list of primary data available for the study area.**

| Primary data | Resolution | Source |
|---|---|---|
| Geological map | 1:50,000 | GSI, Bhukosh, accessed in October 2019 |
| Structural map | 1:50,000 | GSI, Bhukosh, accessed in October 2019 |
| Magnetic TMI | 75 m | GSI, Bhukosh, accessed in October 2019 |
| Ground Gravity | 10,000 m | GSI, Bhukosh, accessed in October 2019 |
| Satellite sensed gravity anomaly and topographical data | 1,600 m | Smith and Sandwell, 1997; Sandwell and Smith, 2009; Sandwell et al., 2013, 2014; accessed in January 2020 |
| SRTM topography | 954 m | Geosoft seeker; accessed in October 2019 |
| Lineaments from remote sensing data | 1:250,000 | GSI, Bhukosh, accessed in October 2019 |
| Known carbonatite occurrences | | Literature review; Table 1 |
| Known prospects | | GSI and AMD, 2020 |

The methodology flow chart is shown in Figure 2.

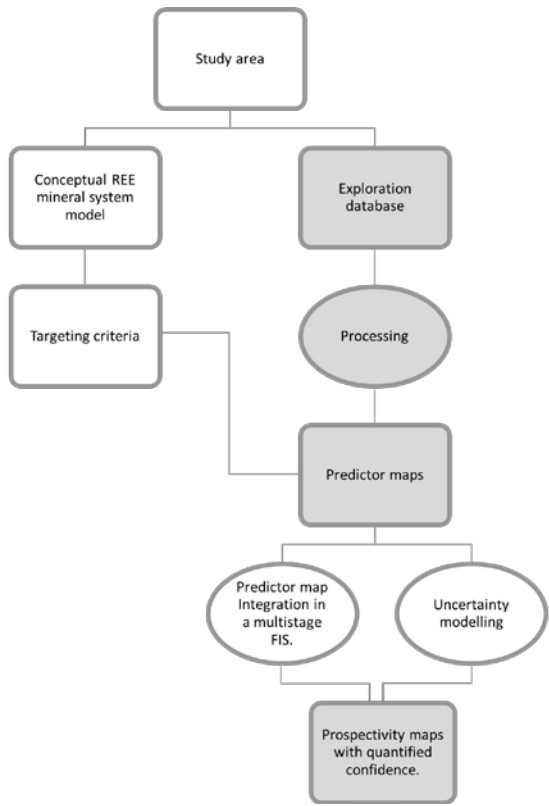

**Figure 2: Flow chart depicting the methodology. Rectangular boxes contain generated objects, and oval boxes contain processes used for creating the objects. Shaded boxes indicate the objects and processes created and implemented in a GIS, respectively.**



The methodology is described in detail in the following subsections.

### 3.1 Mineral systems model for carbonatite-alkaline complex related REE deposits

In this study, we used the generalised conceptual model of carbonatite-alkaline-complex-related REE mineral systems developed by Aranha et al. (in review) based on the framework proposed by McCuaig and Hronsky (2014). Figure 3 illustrates the main features of the model. The main components of the mineral systems are compiled in Table 3 and briefly summarised in the following paragraphs.

**Geodynamic setting:** Carbonatite-alkaline complexes and related REE deposits generally occur in extensional intra-continental rifts and large igneous provinces (LIPs) (#4, 5, 6 in Fig. 3 and Table 3; Woolley and Kjarsgaard 2008a; Woolley and Bailey, 2012; Pirajno, 2015; Simandl and Paradis, 2018). Extensional tectonic settings and associated LIPs are manifestations of mantle plumes (Simonetti et al., 1995, 1998; Bell and Tilton, 2002; Bell and Simonetti, 2010; Ernst and Bell, 2010), which also induce metasomatism of the SCLM, fertile source regions of, and favourable geodynamic settings for, REE deposits related to carbonatite-alkaline complex are interlinked.

**Architecture**: Carbonatite-alkaline complexes and related REE mineral systems derive fluids from the SCLM through large-scale permeable networks of trans-lithospheric structures. Most carbonatite-alkaline complexes are found spatially associated with crustal-scale faults, rifts and shear zones at regional scales (Ernst and Bell, 2010; Woolley and Bailey, 2012; Pirajno, 2015; Simandl and Paradis, 2018; Spandler et al., 2020). Therefore, lithosphere-scale structures form favourable plumbing structures for carbonatite-alkaline-complex-related REE deposits (#7 in Fig. 3 and Table 3). Upper crustal faults, shallow discontinuity structures and joints serve as pathways for focussing fluids to near-surface levels and also form structural traps (#8 in Fig. 3 and Table 3; Ernst and Bell, 2010; Skirrow et al., 2013; Jaireth et al., 2014).

The crystallisation of carbonatites and alkaline complexes along with reactions with the country-rock to form Ca and Mg silicates is accompanied by the removal of $CO_2$, dissolved P and F (Skirrow et al., 2013; Jaireth et al., 2014). The above reactions may cause enrichment of incompatible elements such as REEs, U, Th, Nb, Ba, Sr, Zr, Mn, Fe, Ti (#10, 13, 14, 15, 16, 17, 18, 19 in Fig. 3 and Table 3; Cordeiro et al., 2010; Skirrow et at., 2013; Jaireth et al., 2014; Pirajno, 2015; Mitchell, 2015; Chakhmouradian et al., 2015; Stoppa et al., 2016; Poletti et al., 2016; Giovannini et al., 2017; Simandl and Paradis, 2018; Spandler et al., 2020). Carbonatite-alkaline complexes are often enriched in ferromagnesian minerals that cause well-defined magnetic and gravity anomalies (#9 in Fig. 3 and Table 3; Gunn and Dentith, 1997; Thomas et al., 2016). Fenitisation often enriches country rocks in K and Na (#12 in Fig. 3 and Table 3; Le Bas, 2008; Elliott et al., 2018).

Rare earth element mineralisation in the carbonatites can be in the form of primary REE-bearing minerals (e.g., Mountain Pass, Mariano, 1989; Castor, 2008; Verplanck and Van Gosen, 2011; Van Gosen et al., 2017) or by the precipitation from hydrothermal or late magmatic fluid phases expelled from the carbonatite magmas (Verplanck and Van Gosen, 2011; Skirrow et at., 2013; Jaireth et al., 2014; Van Gosen et al., 2017). Primary REE-bearing cumulates include perovskite, pyrochlore, apatite and calcite, while late-stage REE-bearing minerals include bastnäsite, parasite, and synchysite (#24 in Table 3; Verplanck and Van Gosen, 2011; Skirrow et al., 2013; Van Gosen et al., 2017).





**Figure 3: Idealised genetic model of a carbonatite-alkaline-complex-related REE mineral system (adapted from Aranha et al., under review) cross-referenced to processes listed in Table 3 through the numbers in blue. (A) Depicts the fertility and geodynamic setting along with the plumbing architecture on a regional scale. B, C and D focus on the emplacement architecture at the camp-to-prospect-scale. (B) Shows the idealised geometry of the intrusion and the relation of carbonatites and associated alkaline rocks and fenitisation (C) Presents the near-surface structural architecture and the spatial distribution of associated. (D) Displays the idealised geometry of a carbonatite-alkaline intrusion and the relationship between the magma chamber, ring dykes, cone sheets, and radial dykes.**

180



**Table 3: Conceptual REE mineral systems model (adapted from Aranha et al., under review). The index numbers correlate to the numbers in blue in Fig. 3.**

185

| Setting/process | # | Targeting criteria | Spatial proxies |
|---|---|---|---|
| *Fertility* | | | |
| Mantle metasomatism and low degree partial melting | 1 | Subduction of crust | Subduction zones throughout geological history |
| | 2 | Decompressional melting of mantle and crust due to rifting (crustal thinning) | Rift zones |
| | 3 | Metasomatism driven by a rising mantle plume | Trace of mantle plume |
| *Geodynamic setting and triggers* | | | |
| Continental rifts (Rising Mantle plume) | 4 | Trace of mantle plumes based on plate tectonics through indicative magmatism | Trace of mantle plume through time |
| | 5 | | LIP |
| | 6 | Major global tectonic events - super continental breakups | Plate reconstruction models - rifting |
| *Architecture-plumbing* | | | |
| Migration of magma along existing or new architecture | 7 | Crustal scale discontinuities | Rift structure |
| | | | Deep crust penetrating faults |
| *Architecture-Emplacement* | | | |
| Magma emplacement under structural traps | 8 | Near-surface network of faults | Shallow intersecting faults |
| Carbonatite magma emplacement - Concentration of minerals with a strong magnetic response and contrasting density from the country rocks Concentration of incompatible radioactive elements Hosted by or strongly associated with Ca or Mg carbonate rocks (Carbonatites) | 9 | Anomalous signatures in geophysical data | Anomalous signatures in magnetic and gravity data |
| | 10 | High radioactivity due to U and Th enrichment | High response in radiometric maps due to U and Th Anomalous signatures in geochemical data |
| | 11 | High concentrations of Ca and Mg | Anomalous signatures in geochemical data. |
| Sodic and potassic fenitisation | 12 | Enrichment of K and Na in the surrounding rocks | High response in radiometric maps due to K. |
| Emplacement of incompatible elements in primary carbonatite or secondary carbonatitic veins | 13 | Enrichment of REEs | Anomalous signatures in geochemical data. |
| | 14 | $P_2O_5$, | |
| | 15 | F, Cl and $CO_3$; | |
| | 16 | Nb | |
| | 17 | Ba, Sr, Zr | |
| | 18 | Mn | |
| | 19 | Ti | |





| | | | |
|---|---|---|---|
| Biogeochemical indicators: Absorption of REEs and related elements by plants growing over a potential deposit | 20 | Abundance of Ba, Sr, P, Cu, Co, La, Ce, Pr, Nd, Sm, Dy, Fe, Nb, Ta, U and Y against the background value in the leaves and twigs of the plants and in the Humus. | Plant/Humus anomaly maps |
| Selective absorption of specific wavelengths of the Electromagnetic spectrum | 21 | Characteristic absorption features in remotely sensed images | REE Concentration maps derived from remotely sensed images |
| Carbonatites are commonly spatially associated with alkaline silicate (85%; Woolley and Kjarsgaard, 2008a, b) and in some cases with ultramafic and felsic silicate igneous rocks | 22 | Known alkaline intrusions | Mapped intrusions in geological maps |
| Concentric zoning of carbonate rocks along with magnetic minerals (magnetite) | 23 | Circular outline | Circular features in topographic and geophysical data |
| Variation in mineralogy in REE-bearing minerals and associated alkaline suite of rocks are indicators of emplacement depth as well as erosional level and, therefore, mineralisation potential | 24 | Variation in rock units of the alkaline rock suite and/or Variation of REE minerals | Individual rock and mineralogical units in detailed lithological and mineralogical profiles |

## 3.2 Targeting criteria and predictor maps

The above conceptual model for carbonatite-alkaline-related REE mineral systems was translated into a "targeting model", which is a compilation of processes whose responses can be mapped directly or indirectly in the publicly available datasets for the study area listed in Table 2. The targeting model was used to identify regional-scale mappable targeting criteria for REE deposits in the study area (Tables 4A, B and C).

The mappable targeting criteria for REE deposits in the study area were represented in the form of GIS layers or predictor maps for inputting into the FIS. The details of the primary data, the algorithms and GIS tools and techniques used to generate input predictor maps are provided in Tables 4A, B and C.





**Table 4A: Targeting model, spatial proxies and steps used to derive the predictor maps of the fertility and geodynamic setting components of the REE mineral system in Northwestern India.**

| SNO | Primary Data | Spatial proxy | Individual predictor maps | Procedures used to generate the predictor map | Rationale |
|---|---|---|---|---|---|
| 1 | Magnetic data and 50000 scale structural map of GSI | Barmer Rift/mantle plume trace | Proximity to Rift/trace of mantle plume | Rift outline traced from the vertical derivative of RTP magnetic data and further extrapolated using lineament map outside the coverage of magnetic data. Euclidean distance was calculated to this trace. | Rift can represent the trace of the mantle plume under the crust as a result of crustal extension caused by the rising plume. It also marks a zone of extension and deep permeable faults that facilitate magma flow. Vertical derivative of magnetic data reveal responses from near-surface sources (Gönenç, 2014) where the rift zone is at its maximum dimension. Upwards continued geophysical data shows the extension of the rift at depth. The Barmer rift is assumed to be the trace of the mantle plume. |
| 2 | 50000 scale geology map of GSI | Deccan Large Igneous Province | Proximity to the Deccan Large Igneous Province | Geology map queried and filtered for Deccan large igneous province and then extracted, followed by calculation of Euclidean distance to these extracted features. | Mantle plumes result in the formation of large igneous provinces, and thus, LIPs can demarcate the zone of influence of the mantle plume. Carbonatites are known to be associated with mantle plumes. |

### 3.3 FIS-based prospectivity modelling

The predictor maps were integrated using FIS to generate REE prospectivity maps of the study area. The theoretical exposition of the FIS-based modelling approach and implementation for mineral prospectivity modelling is provided by Porwal et al. (2015) and Chudasama et al. (2016).

The modelling was implemented in the following steps.

1. **Fuzzification of numeric predictor maps**: In the first step, all numeric predictor maps (e.g., the predictor map showing distance to structural lineaments) were converted into fuzzy predictor maps (e.g., proximity to structural lineaments) using membership functions such as linear, piece-wise linear (trapezoidal) or Gaussian (Table 5). However, the output fuzzy membership values of a predictor map are dependent on the parameters of the function used (e.g., mean and standard deviation for Gaussian functions and slope and intercept for linear functions).

Because there were no training data (that is, known deposits) for optimising the fuzzy membership functions, we quantified uncertainty arising from using sub-optimal function parameters (termed "systemic uncertainty"; Porwal et al., 2003b; Lisitsin et al., 2014). The Monte-Carlo-simulation-based algorithm described by Lisitsin et al. (2014) and Chudasama et al. (2017) was used to estimate model uncertainties. Instead of using point values for each function parameter, we used a beta distribution of values and then used a series of Monte Carlo simulations to estimate the function parameter at 10%, 50% and 90% probability levels. The beta distribution was used because it is a bounded distribution that is generally recommended when no training data are available and relies only on expert knowledge about the optimistic, most likely and pessimistic values (Johnson et al., 1995). Three fuzzy maps were generated at 10%, 50%, and 90% probability levels for each predictor map through this step.

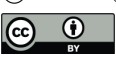
**Table 4B: Targeting model, spatial proxies and steps used to derive the predictor maps of the plumbing pathways architecture component of the REE mineral system in Northwestern In**

| SNO | Primary Data | Spatial proxy | Individual predictor maps | Procedures used to generate the predictor map | Rationale |
|---|---|---|---|---|---|
| 1 | Magnetic RMI data | Lineaments from Magnetic data | 1. Proximity to lineaments derived from magnetic data continued upwards to 2km | RTP magnetic data was continued upwards to 2 km, followed by calculating its total horizontal derivative. Euclidean distance was calculated to the lineaments extracted from the image after edge-enhancements. | Crustal scale structures such as shear zones and crust penetrating faults are excellent conduits for magma transportation. Such features manifest as linear trends on magnetic data (Porwal et al., 2006). Magnetic data continued upwards to 2 km show responses from slightly deeper crustal sources (Jacobsen, 1987; Pawlowski, 1995); thus, these lineaments are perceived to continue to these levels. |
| | | | 2. Proximity to lineaments derived from magnetic data continued upwards to 5km | RTP magnetic data was continued upwards to 5 km, followed by calculating its total horizontal derivative. Euclidean distance was calculated to the lineaments extracted from the image after edge-enhancements. | Crustal scale structures such as shear zones and crust penetrating faults are excellent conduits for magma transportation. Such features manifest as linear trends on magnetic data (Porwal et al., 2006). Magnetic data continued upwards to 5 km show responses from deeper crustal sources (Jacobsen, 1987; Pawlowski, 1995); thus, these lineaments are perceived to continue to these levels. |
| | | | 3. Proximity to lineaments derived from magnetic data continued upwards to 10km | RTP magnetic data was continued upwards to 10 km, followed by calculating its total horizontal derivative. Euclidean distance was calculated to the lineaments extracted from the image after edge-enhancements. | Crustal scale structures such as shear zones and crust penetrating faults are excellent conduits for magma transportation. Such features manifest as linear trends on magnetic data (Porwal et al., 2006). Magnetic data continued upwards to 10 km show responses from further deeper crustal sources (Jacobsen, 1987; Pawlowski, 1995); thus, these lineaments are perceived to continue to these levels. |
| | | | 4. Proximity to lineaments derived from magnetic data continued upwards to 20km. | RTP magnetic data was continued upwards to 20 km, followed by calculating its total horizontal derivative. Euclidean distance was calculated to the lineaments extracted from the image after edge-enhancements. | Crustal scale structures such as shear zones and crust penetrating faults are excellent conduits for magma transportation. Such features manifest as linear trends on magnetic data (Porwal et al., 2006). Magnetic data continued upwards to 20 km show responses from very deep crustal sources (Jacobsen, 1987; Pawlowski, 1995); thus, these lineaments are perceived to continue to these levels. |
| | Satellite Gravity data | Lineaments from satellite gravity data | 1. Proximity to lineaments derived from satellite gravity data continued upwards to 2km | Satellite gravity data was continued upwards to 2 km, followed by calculating its total horizontal derivative. Euclidean distance was calculated to the lineaments extracted from the image after edge-enhancements. | Crustal scale structures such as shear zones and crust penetrating faults are excellent conduits for magma transportation. Such features manifest as linear trends on gravity data (Porwal et al., 2006). Gravity data continued upwards to 2 km show responses from slightly deeper crustal sources (Jacobsen, 1987; Pawlowski, 1995); thus, these lineaments are perceived to continue to these levels. |
| | | | 2. Proximity to lineaments derived from satellite gravity data continued upwards to 5km | Satellite gravity data was continued upwards to 5 km, followed by calculating its total horizontal derivative. Euclidean distance was calculated to the lineaments extracted from the image after edge-enhancements. | Crustal scale structures such as shear zones and crust penetrating faults are excellent conduits for magma transportation. Such features manifest as linear trends on gravity data (Porwal et al., 2006). Gravity data continued upwards to 5 km show responses from deeper crustal sources (Jacobsen, 1987; Pawlowski, 1995); thus, these lineaments are perceived to continue to these levels. |
| | | | 3. Proximity to lineaments derived from satellite gravity data continued upwards to 10km | Satellite gravity data was continued upwards to 10 km, followed by calculating its total horizontal derivative. Euclidean distance was calculated to the lineaments extracted from the image after edge-enhancements. | Crustal scale structures such as shear zones and crust penetrating faults are excellent conduits for magma transportation. Such features manifest as linear trends on gravity data (Porwal et al., 2006). Gravity data continued upwards to 10 km show responses from further deeper crustal sources (Jacobsen, 1987; Pawlowski, 1995); thus, these lineaments are perceived to continue to these levels. |



| SNO | Primary Data | Spatial proxy | Individual predictor maps | Procedures used to generate the predictor map | Rationale |
|---|---|---|---|---|---|
| 2 | | | 4. Proximity to lineaments derived from satellite gravity data continued upwards to 20km | Satellite gravity data was continued upwards to 20 km, followed by calculating its total horizontal derivative. Euclidean distance was calculated to the lineaments extracted from the image after edge-enhancements. | Crustal scale structures such as shear zones and crust penetrating faults are excellent conduits for magma transportation. Such features manifest as linear trends on gravity data (Porwal et al., 2006). Gravity data continued upwards to 20 km show responses from very deep crustal sources (Jacobsen, 1987; Pawlowski, 1995); thus, these lineaments are perceived to continue to these levels. |
| | | | 1. Proximity to lineaments derived from ground gravity data continued upwards to 2km | Ground gravity data was continued upwards to 2 km, followed by calculating its total horizontal derivative. Euclidean distance was calculated to the lineaments extracted from the image after edge-enhancements. | Crustal scale structures such as shear zones and crust penetrating faults are excellent conduits for magma transportation. Such features manifest as linear trends on gravity data (Porwal et al., 2006). Gravity data continued upwards to 2 km show responses from slightly deeper crustal sources (Jacobsen, 1987; Pawlowski, 1995); thus, these lineaments are perceived to continue to these levels. |
| | | | 2. Proximity to lineaments derived from ground gravity data continued upwards to 5km | Ground gravity data was continued upwards to 5 km, followed by calculating its total horizontal derivative. Euclidean distance was calculated to the lineaments extracted from the image after edge-enhancements. | Crustal scale structures such as shear zones and crust penetrating faults are excellent conduits for magma transportation. Such features manifest as linear trends on gravity data (Porwal et al., 2006). Gravity data continued upwards to 5 km show responses from deeper crustal sources (Jacobsen, 1987; Pawlowski, 1995); thus, these lineaments are perceived to continue to these levels. |
| | Ground Gravity data | Lineaments from ground gravity data | 3. Proximity to lineaments derived from ground gravity data continued upwards to 10km | Ground gravity data was continued upwards to 10 km, followed by calculating its total horizontal derivative. Euclidean distance was calculated to the lineaments extracted from the image after edge-enhancements. | Crustal scale structures such as shear zones and crust penetrating faults are excellent conduits for magma transportation. Such features manifest as linear trends on gravity data (Porwal et al., 2006). Gravity data continued upwards to 10 km show responses from further deeper crustal sources (Jacobsen, 1987; Pawlowski, 1995); thus, these lineaments are perceived to continue to these levels. |
| | | | 4. Proximity to lineaments derived from ground gravity data continued upwards to 20km | Ground gravity data was continued upwards to 20 km, followed by calculating its total horizontal derivative. Euclidean distance was calculated to the lineaments extracted from the image after edge-enhancements. | Crustal scale structures such as shear zones and crust penetrating faults are excellent conduits for magma transportation. Such features manifest as linear trends on gravity data (Porwal et al., 2006). Gravity data continued upwards to 20 km show responses from very deep crustal sources (Jacobsen, 1987; Pawlowski, 1995); thus, these lineaments are perceived to continue to these levels. |
| 3 | 50000 scale structural map and 250K scale lineament map of GSI | Lineaments from remote sensing data and inferred faults from the structural map | 1. Proximity to inferred faults and remotely sensed lineaments | Inferred faults and Structural lineaments derived by the GSI were queried, filtered and then extracted out. Euclidean distance was calculated to these lineaments. | Most of the study area is desert covered, where faults are mapped inefficiently. Interpolated lineaments increase the coverage but introduce error. Remote sensing data is a valuable tool to compensate for data gaps. |
| 4 | Magnetic data and 50000 scale structural map of GSI | Barmer Rift | 1. Proximity to Barmer Rift | Rift outline traced from the vertical derivative of RTP magnetic data and further extrapolated using lineament map outside magnetic data coverage. | Rift signify a zone of extension and deep faults for magma flow. Vertical derivative of magnetic data reveal responses from near-surface sources (Gönenç, 2014) where the rift zone is at its maximum dimension. Upwards continued data shows the extension of the rift at depth. |

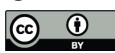



**Table 4C: Targeting model, spatial proxies and steps used to derive the predictor maps of the emplacement architecture component of the REE mineral system in Northwestern India.**

| SNO | Primary Data | Spatial proxy | Individual predictor maps | Procedures used to generate the predictor map | Rationale |
|---|---|---|---|---|---|
| 1 | 50000 scale geology map of GSI | Post-Cambrian non-felsic intrusive bodies | 1. Proximity to younger post-Cambrian, non-felsic intrusives | Geology map was queried and filtered for post-Cambrian, non-felsic rocks and then extracted, followed by calculating Euclidean distance to these extracted features. | Non-felsic intrusions (mainly alkaline intrusions) are associated with carbonatites and represent a magmatic episode that triggered or were part of alkaline and carbonatitic magmatic activity. |
| 2 | | Deccan Large Igneous Province | 1. Proximity to the Deccan Large Igneous Province | Geology map queried and filtered for Deccan large igneous province and then extracted, followed by calculating Euclidean distance to these extracted features. | Mantle plumes result in the formation of large igneous provinces, and thus, LIPs can demarcate the zone of influence of the mantle plume. Carbonatites are known to be associated with mantle plumes. |
| 3 | 250K Lineament map and 50000 scale structural map of GSI | Lineaments from remote sensing data and inferred faults from the structural map | 1. Proximity to inferred faults and remotely sensed lineaments | Inferred faults and Structural lineaments derived by the GSI were queried, filtered and then extracted, followed by calculating Euclidean distance to these extracted lineaments. | Most of the study area is desert covered, where faults are mapped inefficiently. Interpolated lineaments increase the coverage but introduce error. Remote sensing data is a valuable tool to compensate for data gaps. |
| | Magnetic TMI data | Circular features from Magnetic data | 1. Proximity to circular features derived from RTP magnetic data | Total Horizontal derivative was calculated of the RTP image, followed by extraction of circular features after detecting and enhancing circular features. Euclidean distance was calculated to these circular features. | Intrusive carbonatites contain concentric zoning of carbonate rocks along with variable concentrations of magnetite that cause anomalies, ideally concentric or roughly oval in shape (Gunn and Dentith, 1997). The horizontal derivative of RTP magnetic data shows responses from shallow sources; thus, these circular features are considered to be near-surface. |
| | | | 2. Proximity to circular features derived from magnetic data continued upwards to 2km | RTP magnetic data was continued upwards to 2 km, followed by calculating its total horizontal derivative. Euclidean distance was calculated to the circular features extracted from this image after detecting and enhancing circular features. | Intrusive carbonatites contain concentric zoning of carbonate rocks along with variable concentrations of magnetite that cause anomalies, ideally concentric or roughly oval in shape (Gunn and Dentith, 1997). Magnetic data continued upwards to 2 km show responses from slightly deeper crustal sources (Jacobsen, 1987; Pawlowski, 1995); thus, these circular features are perceived to continue to these levels. |
| | | | 3. Proximity to circular features derived from magnetic data continued upwards to 5km | RTP magnetic data was continued upwards to 5 km, followed by calculating its total horizontal derivative. Euclidean distance was calculated to the circular features extracted from this image after detecting and enhancing circular features. | Intrusive carbonatites contain concentric zoning of carbonate rocks along with variable concentrations of magnetite that cause anomalies, ideally concentric or roughly oval in shape (Gunn and Dentith, 1997). Magnetic data continued upwards to 5 km show responses from deeper crustal sources (Jacobsen, 1987; Pawlowski, 1995); thus, these circular features are perceived to continue to these levels. |
| 4 | Satellite Gravity data | Circular features from satellite gravity data | 1. Proximity to circular features derived from gravity data | Total Horizontal derivative calculated of satellite gravity data, followed by extraction of circular features after detecting and enhancing circular features. Euclidean distance was calculated to these circular features. | Intrusive carbonatites contain concentric zoning of carbonate rocks along with variable concentrations of magnetite that cause anomalies, ideally concentric or roughly oval in shape (Gunn and Dentith, 1997). Horizontal derivative of gravity data shows responses from shallow sources; thus, these circular features are considered to be near-surface. |
| | | | 2. Proximity to circular features derived from gravity data continued upwards to 2km | Satellite gravity data was continued upwards to 2 km, followed by calculating its total horizontal derivative. Euclidean distance was calculated to the circular features extracted from this image after detecting and enhancing circular features. | Intrusive carbonatites contain concentric zoning of carbonate rocks along with variable concentrations of magnetite that cause anomalies, ideally concentric or roughly oval in shape (Gunn and Dentith, 1997). Gravity data continued upwards to 2 km show responses from slightly deeper crustal sources (Jacobsen, 1987; Pawlowski, 1995); thus, these circular features are perceived to continue to these levels. |

225





| SNO | Primary Data | Spatial proxy | Individual predictor maps | Procedures used to generate the predictor map | Rationale |
|---|---|---|---|---|---|
| | | | 3. Proximity to circular features derived from gravity data continued upwards to 5km | Satellite gravity data was continued upwards to 5 km, followed by calculating its total horizontal derivative. Euclidean distance was calculated to the circular features extracted from this image after detecting and enhancing circular features. | Intrusive carbonatites contain concentric zoning of carbonate rocks along with variable concentrations of magnetite that cause anomalies, ideally concentric or roughly oval in shape (Gunn and Dentith, 1997). Gravity data continued upwards to 5 km show responses from deeper crustal sources (Jacobsen, 1987; Pawlowski, 1995); thus, these circular features are perceived to continue to these levels. |
| | SRTM Topographic data | Circular features from topographic data | 1. Proximity to circular features derived from topographic data | Circular features were extracted from topographic data after detecting and enhancing circular features. | Topographic data map the surface morphology and relief of the ground. Exposed carbonatite-alkaline ring complexes typically exhibit a circular outline on the surface that is well captured in topographic data. |
| | UCSD Topographic data | Circular features from topographic data | 1. Proximity to circular features derived from topographic data | Circular features were extracted from topographic data after detecting and enhancing circular features. | Topographic data map the surface morphology and relief of the ground. Exposed carbonatite-alkaline ring complexes typically exhibit a circular outline on the surface that is well captured in topographic data. |
| 5 | Magnetic TMI data | Shallow Lineaments from geophysical data | 1. Proximity to surficial lineaments derived from magnetic data | Euclidean distance was calculated to lineaments extracted from the vertical derivative of RTP magnetic data after edge-enhancements. | Shallow, higher-order, local faults and joints aid in focussing the fluids to near-surface areas and can also serve as structural traps. Such features manifest as linear trends on geophysical data (Porval et al. 2006). Vertical derivative of magnetic data reveal responses from near-surface sources (Gönenç, 2014); thus, these lineaments are considered to be near-surface. |
| | Satellite Gravity data | | 2. Proximity to surficial lineaments derived from satellite gravity data | Euclidean distance was calculated to lineaments extracted from the vertical derivative of satellite gravity data after edge-enhancements. | |
| | Ground Gravity data | | 3. Proximity to surficial lineaments derived from ground gravity data | Euclidean distance was calculated to lineaments extracted from the vertical derivative of ground gravity data after edge-enhancements. | |
| 6 | Magnetic TMI data | Points of intersections of shallow lineaments derived from geophysical data | 1. Proximity to intersections of surficial lineaments derived from magnetic data | Points of intersections of lineaments extracted from the vertical derivative of RTP magnetic data were extracted out, followed by calculating Euclidean distance to these points. | Intersections of near-surface lineaments can serve as structural traps. |
| | Satellite Gravity data | | 2. Proximity to intersections of surficial lineaments derived from satellite gravity data | Points of intersections of lineaments extracted from the vertical derivative of satellite gravity data were extracted out, followed by calculating Euclidean distance to these points. | Intersections of near-surface lineaments can serve as structural traps. |
| | Ground Gravity data | | 3. Proximity to intersections of surficial lineaments derived from ground gravity data | Points of intersections of lineaments extracted from the vertical derivative of ground gravity data were extracted out, followed by calculating Euclidean distance to these points. | Intersections of near-surface lineaments can serve as structural traps. |
| 7 | Magnetic TMI data | High magnetic anomalies | 1. Magnetic anomaly map | Analytical signal of the TMI data was calculated to exaggerate anomalous signatures | Carbonatites are often enriched in magnetic minerals such as magnetite that exhibit high magnetic susceptibility. Analytical signal is very useful for identifying magnetic anomalies at lower magnetic latitudes (Rajagopalan, 2003; Keating and Sailhac, 2004). |





**Table 5: Input variables, linguistic values and types of membership functions**

| Input Variable (Spatial Proxy) | Linguistic Values | Type of Membership Function |
|---|---|---|
| **Premise variables** | | |
| · FERTILITY/GEODYNAMIC SETTING | | |
| 1. Proximity to Barmer rift/Plume head. | Proximal, Intermediate, Distal | Piece-wise linear (Trapezoidal)[1], Gaussian[2], Piece-wise linear (Trapezoidal)[3] |
| 2. Proximity to Deccan Large Igneous Province. | Proximal, Intermediate, Distal | Piece-wise linear (Trapezoidal)[1], Gaussian[2], Piece-wise linear (Trapezoidal)[3] |
| · ARCHITECTURE - LITHOSPHERIC PATHWAYS | | |
| 3. Proximity to lineaments derived from magnetic data. | Proximal, Intermediate, Distal | Piece-wise linear (Trapezoidal)[1], Gaussian[2], Piece-wise linear (Trapezoidal)[3] |
| 4. Proximity to Barmer rift. | Proximal, Intermediate, Distal | Piece-wise linear (Trapezoidal)[1], Gaussian[2], Piece-wise linear (Trapezoidal)[3] |
| 5. Proximity to inferred faults and remotely sensed lineaments. | Proximal, Intermediate, Distal | Piece-wise linear (Trapezoidal)[1], Gaussian[2], Piece-wise linear (Trapezoidal)[3] |
| 6. Proximity to lineaments derived from gravity data. | Proximal, Intermediate, Distal | Piece-wise linear (Trapezoidal)[1], Gaussian[2], Piece-wise linear (Trapezoidal)[3] |
| · ARCHITECTURE - EMPLACEMENT | | |
| 7. Proximity to post-Cambrian, non-felsic intrusions. | Proximal, Intermediate, Distal | Piece-wise linear (Trapezoidal)[1], Gaussian[2], Piece-wise linear (Trapezoidal)[3] |
| 8. Proximity to Deccan Large Igneous Province. | Proximal, Intermediate, Distal | Piece-wise linear (Trapezoidal)[1], Gaussian[2], Piece-wise linear (Trapezoidal)[3] |
| 9. Proximity to inferred faults and remotely sensed lineaments. | Proximal, Intermediate, Distal | Piece-wise linear (Trapezoidal)[1], Gaussian[2], Piece-wise linear (Trapezoidal)[3] |
| 10. Cumulative map of Proximity to circular features. | Proximal, Intermediate, Distal | Piece-wise linear (Trapezoidal)[1], Gaussian[2], Piece-wise linear (Trapezoidal)[3] |
| 11. Proximity to surficial faults. | Proximal, Intermediate, Distal | Piece-wise linear (Trapezoidal)[1], Gaussian[2], Piece-wise linear (Trapezoidal)[3] |
| 12. Proximity to intersections of surficial faults. | Proximal, Intermediate, Distal | Piece-wise linear (Trapezoidal)[1], Gaussian[2], Piece-wise linear (Trapezoidal)[3] |
| 13. Geophysical anomaly map. | High, Intermediate, Low | Piece-wise linear (Trapezoidal)[4], Gaussian[5], Piece-wise linear (Trapezoidal)[6] |
| **Consequent Variables** | | |
| · Fertility and Geodynamic setting Potential | High, Intermediate, Low. | Linear[7], Gaussian[7], Linear[7] |
| · Architecture - Pathways prospectivity | High, Intermediate, Low. | Linear[7], Gaussian[7], Linear[7] |
| · Architecture - Emplacement prospectivity | High, Intermediate, Low. | Linear[7], Gaussian[7], Linear[7] |

1 A piece-wise linear (trapezoid) function allots equal weightage (horizontal line section) to areas lying in very close proximity to the input variables while the influence decreases linearly (inclined line section) as the distance increases. Such a function suits well to represent close proximity relations. For instance, close proximity to faults can be described as the first few kilometres being surely proximal and are assigned the fuzzy membership value of 1. After a certain threshold, the level of a given distance being proximal decreases progressively; the fuzzy membership value linearly decreases until it reaches zero.

2 The uncertainty is associated with the determination of intermediate proximity levels is much higher as a subjective value of intermediateness is estimated based on expert knowledge. The membership values reduce gradually as we move away from this estimated distance value. A Gaussian function best represents such a relation since the 'bell-shape' allots high weightage to the estimate values and its immediate surroundings.

3 Beyond a certain threshold distance, the input variable is considered to have no geological influence on mineralisation and can be assigned an equal weightage of being distal (horizontal line section of the trapezoidal function). The weightage would increase steadily in a linear manner as this threshold is approached (inclined line section of the trapezoidal function). Hence, a piece-wise linear function was used to represent distal relationships.

4 A piece-wise linear (trapezoid) function allots equal weightage (horizontal line section) to values beyond an estimated threshold to represent high anomalous values. The threshold is such that values beyond it would surely be anomalously high. The weightage decreases linearly as the geophysical anomaly values reduce from the estimated threshold (inclined line section). Accordingly, a piece-wise linear function was used to represent high geophysical anomaly values.

5 Magnetic susceptibility generally conforms to a log-normal distribution (Latham et al., 1989). Therefore, a gaussian function was used to represent intermediate values.

6 Equal weightage of 'low-ness' (horizontal line section of the trapezoidal function) was allotted to values that were considered to be too low to be indicative of REE mineralisation. The membership values reduce linearly as magnetic susceptibility values increase. Therefore, a piece-wise linear function was used to represent low geophysical anomaly values.

7 The output (consequent) variables have been assigned linear membership functions to model the favourability on a linear scale.

Every membership function described above relies on an assumed or estimated parameter/threshold. The variation of these parameters/thresholds that influence the shape of the membership functions were modelled using Monte Carlo simulations. The degree of variation was represented by a beta (PERT) distribution which is defined by

$$\mu = \frac{a + 4b + c}{6}$$

where a is the minimum limit up to which a given parameter/threshold may vary, b is the most likely value that is estimated based on our knowledge, and c is the maximum variation value.

The values of a and c move further away from b as uncertainty increases.

The value of each parameter/threshold was then simulated 1000 times within the constraints of the assumed beta (PERT) distribution.

These simulated values at three probability levels (10%, 50% and 90%) were used to define the shape of the fuzzy membership functions in separate respective FIS and therefore, determine the fuzzy membership values for each predictor map, at the respective probabilities.




2. **FIS-based prospectivity modelling:** In the second step, a multi-stage FIS was designed to mimic the geological reasoning an exploration geologist for delineating regional-scale exploration targets.

In the first stage, a series of FISs were developed to generate fuzzy prospectivity maps for individual components

of the REE mineral systems by combining their respective fuzzy predictor maps. The FISs for fertility/geodynamic settings, whole lithosphere architecture and near-surface architecture (Fig. 4) comprised 5, 8 and 11 fuzzy if-then rules, respectively, which are shown in Table 6A, 6B and 6C, respectively. Since each predictor map was converted into three fuzzy maps at 10%, 50% and 90% probability levels, the outputs of this step were three fuzzy prospectivity maps for each component at 10%, 50% and 90% probability levels.

In the second stage, the fuzzy prospectivity maps of the individual mineral-system components were combined using the fuzzy product operator (Fig. 4D) to generate three REE prospectivity maps of the study area at 10%, 50% and 90% probability levels.

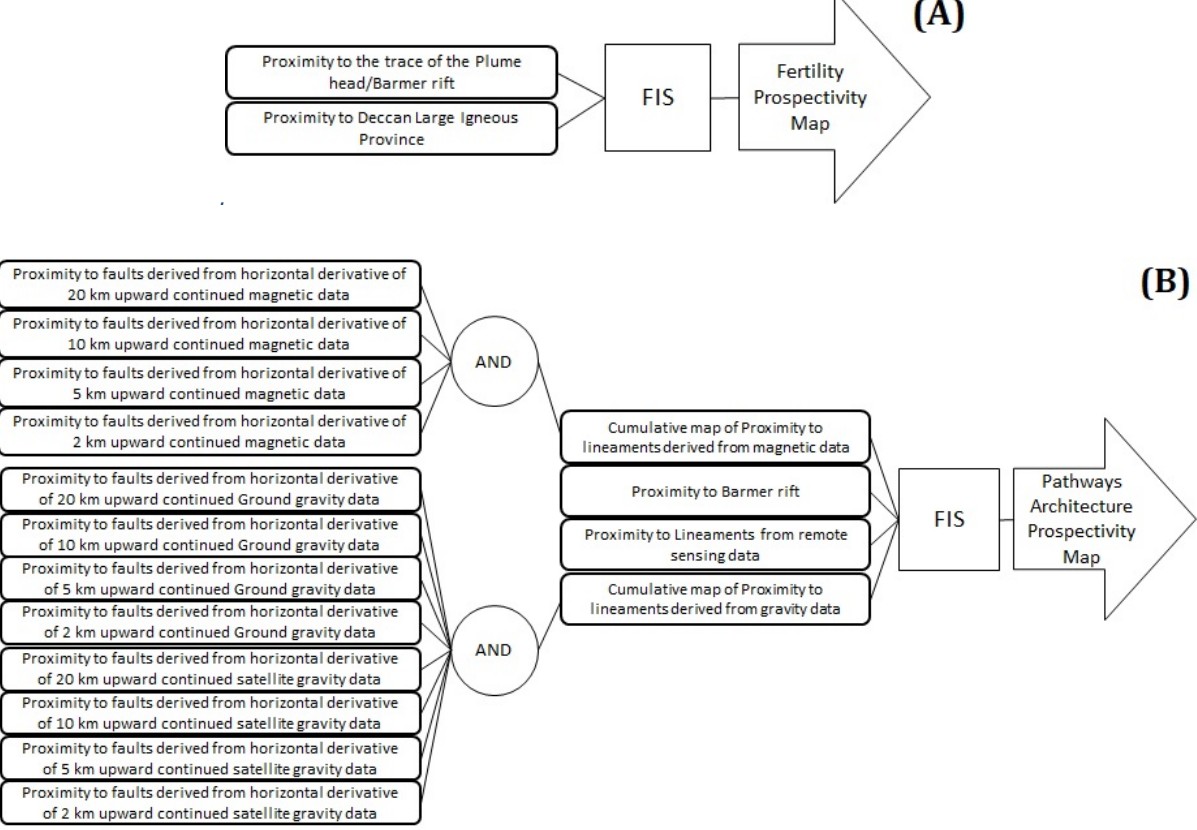







**Figure 4: The multi-stage FIS for REE prospectivity mapping in the study area. (A) FIS for generating fuzzy prospectivity maps for fertile sources and favourable geodynamics settings. (B) FIS for generating fuzzy prospectivity maps for favourable whole lithosphere architecture for transportation of REE-enriched carbonatite-alkaline magma. (C) FIS for generating fuzzy prospectivity maps for favourable shallow crustal (near-surface) architecture for emplacement of carbonatite-alkaline complexes. (D) Second stage FIS combines the above three prospectivity maps obtained from the first stage and generates the final outputs.**

Solid Earth Discussions — Open Access — EGU

**Table 6A: Fuzzy if-then rules used for generating fuzzy prospectivity maps for fertile sources and favourable geodynamics settings.**

| | Consequent (IF part) | | | Antecedent (then part) | |
|---|---|---|---|---|---|
| | IF Plume head/rift is | | and/or LIP is | then Fertility/Geodynamic setting prospectivity is | |
| 1 | proximal | and | proximal | then | High |
| 2 | not distal | and | not distal | then | High |
| 3 | distal | or | distal | then | low |
| 4 | intermediate | or | not distal | then | intermediate |
| 5 | not distal | or | intermediate | then | Intermediate |

**Table 6B: Fuzzy if-then rules used for generating fuzzy prospectivity maps for favourable whole lithosphere architecture for transportation of REE-enriched alkaline-carbonatite magma.**

| | Consequent (IF part) | | | Antecedent (then part) | |
|---|---|---|---|---|---|
| | IF rift is | Lineaments derived from remote sensing are | Lineaments derived from Magnetic data are | Lineaments derived from gravity data are | then Pathways architecture prospectivity is |
| 1 | proximal | and proximal | and proximal | and proximal | then High |
| 2 | intermediate | and proximal | and proximal | and proximal | then High |
| 3 | IF | and proximal | and proximal | and proximal | then High |
| 4 | distal | or distal | or distal | or distal | then low |
| 5 | intermediate | and intermediate | and intermediate | and intermediate | then Intermediate |
| 6 | IF | and proximal | and proximal | and proximal | then High |
| 7 | not distal | and not distal | and not distal | and proximal | then High |
| 8 | not distal | and not distal | and not distal | and proximal | then High |




**Table 6C: Fuzzy if-then rules used for generating fuzzy prospectivity maps for favourable shallow crustal (near-surface) architecture for emplacement of alkaline-carbonatite complexes.**

| # | Consequent (IF part) | Antecedent (then part) |
|---|---|---|
| 1 | **IF** Lineaments derived from remote sensing are proximal **and** intrusions are proximal **and** LIPs are not distal **and** circular features are proximal **and** surficial faults derived from geophysical data are proximal **and** intersections of surficial faults derived from geophysical data are proximal **and** magnetic anomalies are high | **then** Emplacement architecture prospectivity is High |
| 2 | **IF** Lineaments derived from remote sensing are proximal **and** intrusions are not distal **and** LIPs are not distal **and** circular features are proximal **and** surficial faults derived from geophysical data are proximal **and** intersections of surficial faults derived from geophysical data are proximal **and** magnetic anomalies are high | **then** Emplacement architecture prospectivity is High |
| 3 | **IF** Lineaments derived from remote sensing are proximal **and** surficial faults derived from geophysical data are proximal | **then** Emplacement architecture prospectivity is High |
| 4 | **IF** Lineaments derived from remote sensing are distal **or** intrusions are distal **or** LIPs are distal **or** circular features are distal **or** surficial faults derived from geophysical data are distal **or** intersections of surficial faults derived from geophysical data are distal **or** magnetic anomalies are low | **then** Emplacement architecture prospectivity is low |
| 5 | **IF** Lineaments derived from remote sensing are intermediate **and** intrusions are intermediate **and** LIPs are intermediate **and** circular features are intermediate **and** surficial faults derived from geophysical data are intermediate **and** intersections of surficial faults derived from geophysical data are intermediate **and** magnetic anomalies are intermediate | **then** Emplacement architecture prospectivity is Intermediate |
| 6 | **IF** circular features are proximal **and** surficial faults derived from geophysical data are proximal **and** intersections of surficial faults derived from geophysical data are proximal **and** magnetic anomalies are high | **then** Emplacement architecture prospectivity is High |
| 7 | **IF** circular features are distal **and** surficial faults derived from geophysical data are distal **and** intersections of surficial faults derived from geophysical data are distal **and** magnetic anomalies are low | **then** Emplacement architecture prospectivity is low |
| 8 | **IF** circular features are intermediate **and** surficial faults derived from geophysical data are intermediate **and** intersections of surficial faults derived from geophysical data are intermediate **and** magnetic anomalies are intermediate | **then** Emplacement architecture prospectivity is intermediate |
| 9 | **IF** surficial faults derived from geophysical data are proximal **and** intersections of surficial faults derived from geophysical data are proximal **and** magnetic anomalies are high | **then** Emplacement architecture prospectivity is High |
| 10 | **IF** surficial faults derived from geophysical data are intermediate **and** intersections of surficial faults derived from geophysical data are intermediate **and** magnetic anomalies are intermediate | **then** Emplacement architecture prospectivity is intermediate |
| 11 | **IF** surficial faults derived from geophysical data are distal **and** intersections of surficial faults derived from geophysical data are distal **and** magnetic anomalies are low | **then** Emplacement architecture prospectivity is low |





**3.   Generation of confidence map:** In the third step, stochastic uncertainties, which arise from the limitations of public-domain datasets and procedures used for generating the predictor maps, were quantified in terms of confidence values for each predictor map using the techniques described by Porwal et al. (2003b) and Joly et al. (2012). The confidence value for each predictor map was assigned based on the degree of representativeness of the predictor map – that is, how well it represents the mineralisation process it seeks to map. A predictor map

was assigned a high confidence value if it directly mapped the targeting criteria and a low confidence value if it indirectly mapped the response of the targeting criterion. The confidence factor also captured the fidelity and precision of the primary dataset from which the input was derived. The confidence factor for all predictor maps, along with the justifications, are given in Table 7. The output confidence map was generated by combining the confidence factors of different predictor maps using the same fuzzy inference systems that were used for

prospectivity modelling.

**Table 7: Confidence values allotted to each of the predictor maps used in the FIS modelling.**

| Predictor map | Confidence value | Justification |
|---|---|---|
| Proximity to the Deccan Large Igneous Province | 0.9 | LIP mapped extensively on the field at 1:50000 scale. |
| Proximity to the trace of Réunion mantle plume | 0.4 | Interpreted map; the trace of the plume was derived based on the assumption that it coincides roughly with the Barmer-Cambay rift. |
| Proximity to the Barmer Rift | 0.8 | The rift was traced using magnetic data and remotely sensed lineaments and further cross verified with the traces published by Bladon et al. (2015a, b); Dolson et al. (2015). |
| Proximity to lineaments derived from magnetic data | 0.75 | Lineaments were mapped from high-resolution magnetic data. |
| Proximity to lineaments derived from gravity data | 0.7 | Lineaments were mapped from low-resolution gravity data. |
| Proximity to lineaments from remote sensing data and inferred faults from structural maps | 0.5 | Lineaments were mapped from remote sensing data. The faults are inferred, not directly mapped. |
| Proximity to post-Cambrian, non-felsic intrusives | 0.8 | Exposed intrusions directly mapped on field at 1:50000 scale. |
| Proximity to circular features | 0.5 | Circular features were mapped from high-resolution magnetic, low-resolution gravity and topographic data. |
| Proximity to surficial lineaments derived from geophysical data | 0.7 | Lineaments were mapped from high-resolution magnetic and low-resolution gravity data. |
| Proximity to intersections of surficial lineaments derived from geophysical data | 0.7 | Lineaments were mapped from high-resolution magnetic and low-resolution gravity data. |
| Magnetic anomaly map | 0.7 | Anomalies mapped from high-resolution magnetic data. |





Finally, the three REE prospectivity maps of the study area at 10%, 50% and 90% probability levels were blue-to-red colour-
coded and draped over the confidence map for viewing as 3D surface models. In the 3D surface models, the colours represented
prospectivity (blue tones signify low prospectivity and red tones signify high prospectivity), and elevation represented
confidence (depressions signify low confidence and elevations signify high confidence).

## 4 Results

The final outputs are shown as continuous-scale (relative) prospectivity maps at 10%, 50% and 90% probability levels draped
over confidence map in Figures 5 A, B, and C. High prospectivity areas cluster around the carbonatite occurrences of Sarnu-
Dandeli and Kamthai. Several areas to the south of Mundwara and Barmer also show high prospectivity at high probability
levels. In contrast, some areas in the north and northwest of Sarnu-Dandeli show high prospectivity at low probability levels.
Throughout the study area, prospective areas follow the outline of major faults and lineaments. A circular area to the east of
Sarnu-Dandeli shows high prospectivity at low and moderate probabilities; however, it shows low prospectivity at the high
probability level. A small patch south of the circular outline shows high prospectivity across all probability levels.

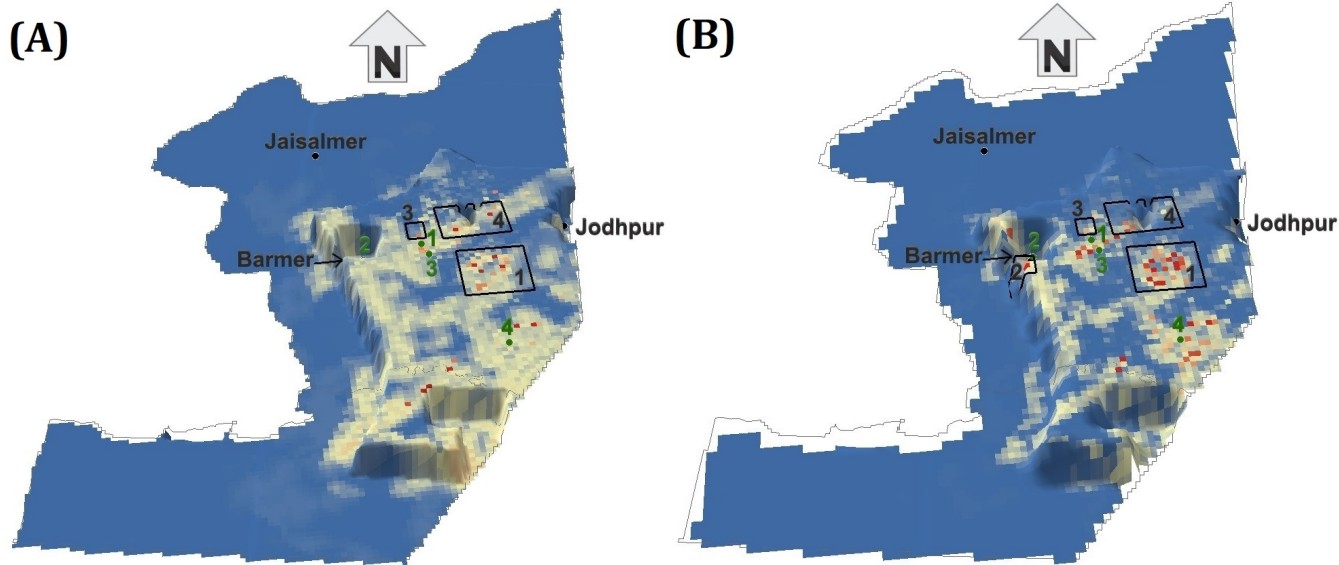





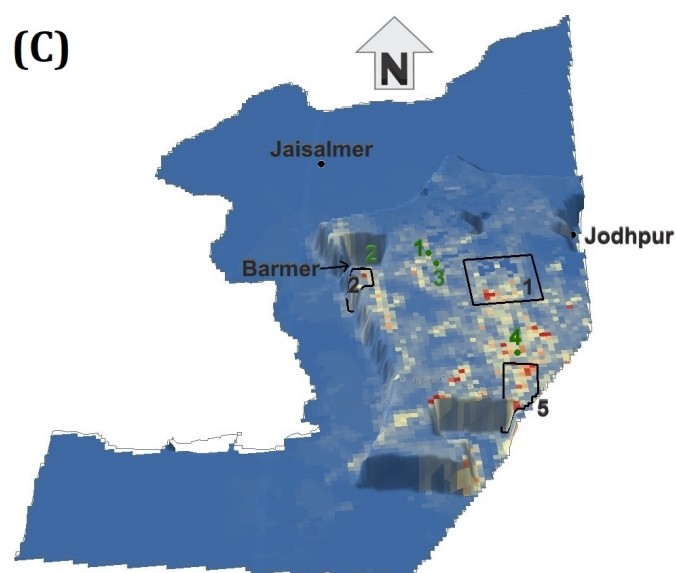

Figure 5: Continuous scale prospectivity maps at 10%, 50% and 90% probability levels draped over the confidence layer, shown in (A), (B) and (C), respectively. The colours mark increasing prospectivity from low (blue) to high (red). The elevations mark high confidence in the data used for prospectivity modelling. Black balls demarcate major cities, and green balls demarcate known carbonatite occurrences; green numbers correspond to the known carbonatite occurrences: 1 – Sarnu Dandeli, 2 – Kamthai, 3 – Danta-Langera-Mahabar, 4- Mundwara. Areas marked with black numbered rectangles are discussed in Section 6.

## 5 Discussion and Recommendations

The lack of known carbonatite-alkaline complexes REE deposits in the study area precluded the use of data-driven approaches, and therefore we opted to apply the knowledge-driven FIS approach. Because FISs are constructed in natural language using simple if-then rules, they are transparent and easy to construct and interpreted by geologists (Porwal et al., 2015). The multi-stage FIS in this study replicates the structure of the REE mineral system model and encapsulates the geological reasoning that an exploration geologist would use to delineate regional-scale exploration targets. The rules utilise fuzzy 'AND' (minimum), 'OR' (maximum) (Bonham-Carter, 1994; Porwal et al., 2015) operators; these operators are used in such a way as to narrow down prospectivity areas as efficiently as possible. Mathematical functions and operators are used to convert the if-then rules in English into machine-readable mathematical values.

In the first stage, the first FIS maps REE fertility and favourable geodynamic settings (Fig. 4A and Table 6A) by delineating areas that are likely to be underlain by plume-metasomatised SCLM. Considering the size of a typical mantle plume, these areas are expected to be very large. The second FIS maps favourable lithospheric architecture for the transportation of REE-enriched carbonatite-alkaline magma (Fig. 4B and Table 6B) and narrows down the target areas identified by the first FIS to areas that are proximal to trans-lithospheric structures. The target areas demarcated by the second FIS are also relatively large





as immense trans-lithospheric structures are expected to have a large zone of influence. The third FIS maps favourable shallow
crustal (near-surface) architecture for the emplacement of carbonatite-alkaline complexes (Fig. 4C and Table 6C) and further
narrows down the target area to camp-size areas that are facilitated by near-surface higher-order structures. These individual
FIS in the first stage rely on simple logic-based rules to integrate the individual predictor maps (Tables 6A, B, and C). The
rules were framed based on our understanding of the REE mineral system. The use of AND operator in the IF parts of the rules
defining high prospectivity ensured that a pixel would get a high prospectivity value only if it is proximal to predictor features
on all predictor maps. Similarly, the use of the OR operator in the IF parts of the rules defining low prospectivity ensured that
a pixel would get a low prospectivity low even if it is distal to predictor features on any one of the predictor maps. As a result,
the extents of the areas with background (low) prospectivity are maximised, and high-prospectivity zones are narrowed down
efficiently.

In the second stage of the multi-stage FIS, the output prospectivity maps of the individual components were integrated using
the fuzzy product operator, which calculates the mathematical product of all input predictor maps (Bonham-Carter, 1994;
Porwal et al., 2015). Since the individual FIS output values range between 0 and 1, it decreases the final integrated prospectivity
values.

We also attempted to quantify the different uncertainties associated with the prospectivity analysis process in this contribution.
Systemic uncertainty arises from the subjective estimation of mathematical parameters that determine the shape of the fuzzy
membership functions used to convert numerical predictor maps to fuzzy predictor maps, which greatly influence the final
prospectivity maps. Instead of point values, Beta-PERT distributions of values were used for the parameters of the fuzzy
membership functions. The parameters of the beta functions (optimistic, most likely and pessimistic values) were assigned
based on a geological evaluation of the decay of the influence of a targeting criteria with distance (Table 5). Monte-Carlo
simulations provided the fuzzy membership values at 10%, 50%, and 90% probability levels, which yielded three sets of fuzzy
predictor maps at 10%, 50%, and 90% probability levels. These three sets of predictor maps were then integrated through
respective multi-stage FIS to obtain the final prospectivity maps at 10%, 50%, and 90% probability levels.

 Stochastic uncertainties were quantified based on the approach described by Porwal et al. (2003b), González-Álvarez et al.
(2010) and Joly et al. (2012) by assigning each predictor map a particular confidence value as per the Sherman-Kent scale
(Jones and Hillis, 2003; Kreuzer et al., 2008). Most previous workers (e.g., Porwal et al., 2003b; González-Álvarez et al., 2010;
Joly et al., 2012) incorporated confidence values in the fuzzy membership values. However, according to the fuzzy set theory,
fuzzy membership value is simply a measure of the strength of an input map as a predictor of the targeted deposit and is
independent of the quality of data used to generate the input predictor map. Therefore, we created separate confidence maps
for all predictor maps and propagated them through the same multi-stage FIS (Fig. 4) to generate an integrated confidence
map.
Conjunctive interpretations of prospectivity maps and confidence maps can help in making decisions regarding follow up
exploration. In the present study, we used the matrix shown in Table 8 to recommend follow-up exploration.



**Table 8: Matrix summarising the target areas quantified according to probability and confidence levels and further exploration recommended for the identified targets.**

| Target | Prospectivity | Probability | Confidence | Interpretation | Recommendation |
|---|---|---|---|---|---|
| Known carbonatite occurrences of Mundwara, Sarnu-Dandeli and Kamthai and several patches surrounding them | High | High | High | High prospectivity because of the possible presence of extended arms of the central carbonatite-alkaline complex intrusion | Apply direct detection techniques such as high-resolution air-borne radiometric surveys and drilling to identify mineral deposits. |
| (1) Circular outline east of Sarnu-Dandeli (Fig. 5A and B; rectangle number 1); and a small patch just south of the circular outline (within rectangle 1 in Figs. 5A, B and C). | High | Moderate / High | High | The circular outline represents the Siwana ring intrusion, consisting of alkali granites and rhyolites. High prospectivity may result from the consistent presence of lineaments and magnetic response of the intrusion. | Follow-up detailed exploration using high-resolution air-borne radiometric surveys and ground geochemical sampling of outcrops, especially of the patch south of the Siwana ring complex. |
| (2) Small patch south of Barmer town | High | High | Moderate | High prospectivity because of the intersection of lineaments | Aerial radiometric surveys are recommended, followed by high-resolution ground gravity surveys and later drilling if the radiometric surveys yield positive results. |
| (3) North of the Sarnu-Dandeli carbonatite occurrence | High | Moderate | High | High prospectivity because of the high density of lineaments in this section and high magnetic anomalies | High-resolution ground gravity and aerial radiometric surveys are recommended, followed by ground sampling and drilling if the radiometric and gravity surveys yield positive results. |
| (4) Northeast of the Sarnu-Dandeli carbonatite occurrence | High | Moderate | Moderate | High prospectivity because of the high density of lineaments in this section and high magnetic anomalies | High-resolution ground gravity and aerial radiometric surveys are recommended, followed by ground sampling and drilling if the radiometric and gravity surveys yield positive results. |
| (5) Several areas east and southeast of Mundwara carbonatite occurrence | High | High | High | High prospectivity because of consistent overlap of lineaments derived from each geophysical source | Additional data collection - High-resolution ground gravity, aerial radiometric surveys and geochemical sampling of outcrops to delineate deposits. |


Along with the known carbonatite occurrences of Mundwara, Sarnu-Dandeli and Kamthai, high prospectivity (orange-red colours in Fig. 5A, B and C) is noted at several scattered patches immediately surrounding Sarnu-Dandeli and Mundwara at high probability and confidence levels. These scattered patches can represent scattered arms of the central carbonatite-alkaline complex intrusion. Direct detection studies are recommended in these locations.

At low probability levels (Fig. 5A and B), moderate to high prospectivity is seen over a circular outline east of Sarnu-Dandeli (Fig. 5A and B; rectangle number 1); and also, over a small patch just south of the circular outline (within rectangle 1 in Figs. 5A, B and C). The circular outline corresponds to the Siwana ring intrusion, which consists of alkali granites and rhyolites.



The Siwana ring intrusion is part of the Neoproterozoic Malani LIP (Bhushan and Mohanty, 1988). However, the Siwana ring intrusion shows low prospectivity at high probability (Fig. 5C; rectangle number 1), while the smaller patch to its south

consistently shows high prospectivity at high probability and confidence levels. The high values may be caused by the consistent presence of lineaments in this region and the magnetic response of the intrusion. It is noteworthy that although not a carbonatite-alkaline complex, the peralkaline Siwana ring complex does contain REE potential and has been assessed for REE mineralisation (Bhushan and Somani, 2019). Further detailed assessment of this region is recommended, with detailed radiometric surveys and geochemical sampling, especially of the patch south of the Siwana ring complex that shows high

prospectivity at high probability levels.

A small area south of Barmer shows high prospectivity at high probability and moderate confidence levels (Fig. 5B and C; rectangle number 2). This area exhibits high prospectivity due to the intersection of lineaments. Two more areas to the north and northeast of the Sarnu-Dandeli carbonatite occurrence show high prospectivity at moderate probability and confidence levels (Figs. 5A and B, rectangles 3 and 4, respectively). A high density of lineaments in this section and high magnetic

anomalies are the likely causes. Aerial radiometric surveys are recommended at all three locations, followed by ground sampling and drilling if the radiometric surveys yield positive results.

Several areas east and southeast of Mundwara show high prospectivity at high probability and confidence levels (Fig. 5C; rectangle 5). This is likely due to the consistent overlap of lineaments derived from each geophysical source at these locations. Acquiring additional data would help in delineating the target zone in these areas.

The emplacement of the carbonatite-alkaline complexes in the study area was related to the large-scale rifting and splitting of India from Madagascar and later from Seychelles, which also triggered the Deccan volcanism. A similar mode of origin is envisaged for several other carbonatite-alkaline complexes worldwide. Ernst and Bell (2010) have identified several carbonatite provinces that are emplaced in an extensional setting, associated with a mantle plume and a LIP. These include, along with the Deccan province, the Afar province (East Africa), Paraná-Etendeka (South America and Africa), Siberian

province (Russia), East European Craton-Kola province (Eastern Europe), Central Iapetus province (North America, Greenland and the Baltic region), and Pan-superior province (North America). This paper's methodologies can be used for exploration targeting REEs in these provinces.

Furthermore, at the time of emplacement of these carbonatite-alkaline complexes, the Indian subcontinent was located close to Madagascar and Seychelles. Therefore, similar complexes could occur in Madagascar and Seychelles also. The Barmer rift

is the northern extension of the Cambay rift, which forms a triple junction in western India along with the Kutch rift. Thus, carbonatite-alkaline complexes are also expected along the Cambay rift and Kutch rifts, also possibly along the offshore E-W trending Gop and the NNW-SSE trending West Coast rift zones on the western coast of India. Kala-Dongar (Sen et al., 2016) and Murud-Janjira (Sethna and D'Sa, 1991) are known minor occurrences of carbonatites along the Kutch and West Coast rift zone, respectively. Moreover, the Gop rift is the western extension of the Son-Narmada-Tapti (SONATA) rift zone, along

which several significant occurrences of the Chhota-Udepur carbonatite district are found. A similar study may help in identifying exploration targets for REEs in these regions. Paleo-reconstruction of the geography to the time when these



complexes were being emplaced and analysing the prospectivity of the entire Deccan province (including western India, Madagascar and Seychelles) may help identify more prospective targets for carbonatite related REEs.

## 6 Summary and Conclusions

Rare earth elements comprise of 17 metallic elements that are considered as 'critical metals' for future development of environmentally friendlier and technologically based societies. India's production entirely comes from secondary beach placer deposits on the western and eastern coasts. Even though no primary economic-grade deposit of REE is identified in India, there is significant latent potential for carbonatite-related REE deposits. This study has developed a knowledge-driven, GIS-based prospectivity model for exploration targeting of REEs associated with carbonatite-alkaline complexes in the western

Rajasthan, northwestern India.

The generalised mineral systems model for carbonatite-alkaline complexes related REEs described by Aranha et al. (under review) was used to identify regional-scale targeting criteria for REE in the study area. Several predictor maps were derived from public-domain geological, geophysical and satellite data based on the mineral systems model. A multi-stage FIS was constructed to represent the different components of the mineral system. The first stage of the multi-stage FIS comprises of

three individual FIS to represent (1) plume-metasomatised SCLM in an extensional regime that make up fertile source regions for REE-bearing fluids and favourable geodynamic settings; (2) trans-lithospheric structures that provide favourable lithospheric architecture for the transportation of REE-enriched carbonatite-alkaline magma; and (3) near-surface higher-order structures that make up a shallow crustal architecture facilitating emplacement of carbonatite-alkaline complexes.

Systemic uncertainties associated with the fuzzification of the predictor maps was quantified based on the procedure described

by Lisitsin et al. (2014) and Chudasama et al. (2017) that produced prospectivity maps at 10%, 50% and 90% confidence levels. Stochastic uncertainties associated with the primary data used and the processing methods adopted to derive predictor maps were quantified based on the procedure described by Porwal et al. (2003b), producing a confidence layer over which the prospectivity maps were draped.

Based on the results, a solid structural control over the emplacement of carbonatite-alkaline complexes is recognised. The

following are the recommendations based on the results of this study. Project-scale detailed ground exploration is recommended for the Kamthai-Sarnu-Dandeli and Mundwara regions and their immediate surroundings, where patches of high prospectivity are recorded at high probability levels. More data collection is recommended for the Siwana ring complex, particularly for the high prospectivity region found to its immediate south. Similarly, high resolution data should be collected in the regions to the north and northeast of Sarnu-Dandeli, south of Barmer, and the south of Mundwara to better resolve and

delineate targets for ground exploration.

The prospectivity-analysis workflow presented in this paper can be applied to other geodynamically similar regions globally for targeting geological provinces for follow-up exploration such as the Deccan province, the Afar province (East Africa), Paraná-Etendeka (South America and Africa), Siberian province (Russia), East European Craton-Kola province (Eastern





Europe), Central Iapetus province (North America, Greenland and the Baltic region), and Pan-superior province (North
America).

*Code/Data availability:* NA

*Author contribution:* Conceptualisation, M.A. and A.P.; methodology, M.A., A.P. and M.S.; software, M.A. and M.S.; formal
analysis, M.A., A.P. and I.G.; investigation, M.A., A.P. and I.G.; resources, M.A., A.P., A.M., and K.R.; data curation, M.A.,
M.S. and A.M.; writing, M.A., A.P., and I.G.

*Competing interests:* The authors declare no conflict of interest.

*Acknowledgements:* This paper benefited greatly from valuable discussions with Dr Majid Keykhay-Hosseinpoor and Dr Bijal
Chudasama, for which they are thanked.

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
