# Peer review of "REEs associated with carbonatite-alkaline complexes in western Rajasthan, India: exploration targeting at regional-scale"

_Solid Earth, 2021_

## Author Comment (AC1)

This looks like a great paper and more data-driven methods can definitely benefit the relative young science of exporation targeting for REE deposits, particularly those related to alkaline and carbonatite complexes. Looking forward to reading it when it is published! I do have some minor comments and corrections regarding the geochemical and petrological aspects of this paper.

We are very grateful for the encouraging remarks and welcome the constructive comments. Please see our responses to the comments in blue belowline 32: Although IUPAC includes this entire list, Pm is an unstable element and for practical reasons in geology does not exist. It is better to remove it. Scandium also probably does not belong here, as it is not an element that is generally associated with the other REEs, and it not sourced from carbonatites and alkaline complexes.

**Response:** We appreciate your suggestion and agree that Pm is an unstable element, not found in nature and that Sc is generally not associated with other REEs sourced from carbonatites and alkaline complexes. However, they were included in the list as the IUPAC recognised list was being quoted. To implement this aspect, we will, however, add the statement in the revised version: "However, since Pm is an unstable element and Sc is not an element sourced from carbonatites-alkaline complexes, this study does not claim to have these elements as part of the targeting outputs." in line 94 to make the reader aware of this.

line 45: Placers are the major source in India, but not globally. The largest operating REE mine is Bayan Obo, which is indeed a high temperature carbonatite.

**Response:** We acknowledge the error and thank you for pointing it out, we intend to rephrase the sentence in line 44 and 45 to "Although the majority of **Indian** production of REEs comes from low-temperature deposits such as regolith-hosted and heavy-mineral placer (IBM yearbook 2018, 2019), the bulk of geological resources are in high-temperature magmatic deposits, particularly those associated with carbonatites (e.g., Bayan Obo, Inner Mongolia, China; Mount Weld, Western Australia; Maoniuping, South China; Mountain Pass, USA etc.; Gonzalez-Alvarez et al., 2021 and references therein)."

lines 62-73: Is a list of all approaches necessary? If yes, then I think it would look better in a table. It is challenging to read as it is now, in paragraph text.

**Response:** We agree with your suggestion and plan to remove the paragraph in the next revision.

line 91: You're saying that "no deposit has been identified in the province so far", but in Table 1 you're showing Kamthai, which has a delineated resource estimate?

**Response:** You are very right. Thank you for flagging this. During the next revision, the sentence in line 91 will be rephrased as "Although a well-established carbonatite province that is widely considered prospective for REE deposits, **just a single** deposit has been identified in the province so far."

line 165: Reactions between carbonatites and country rocks are very common in carbonatites, and it is correct that these reaction remove CO2. However, it has very little to do with the enrichment in REE. If anything, it allows REE to be deposited into REE-rich silicate minerals, essentially removing REE from the melt. This has been discussed in detail in our own work (Anenburg & Mavrogenes 2018, available https://doi.org/10.2475/03.2018.03 or open access at http://hdl.handle.net/1885/143148)
REE are enriched in carbonatites primarily because they are incompatible elements, and Na+K in the melt allow them to be soluble all the way down to the final stages of magmatic fractionation, causing their enrichment in the last batches of carbonatite melt. See our work (Anenburg et al 2020 available open access at https://doi.org/10.1126/sciadv.abb6570 )

**Response:** We thank you for the suggestions and new references provided. This input has greatly improved the model. The paragraph (lines 165 – 175) will be modified as follows in the upcoming revision to adapt the new references –

*"The crystallisation of carbonatites and alkaline complexes along with reactions with the country-rock to form Ca and Mg silicates is accompanied by the removal of $CO_2$, dissolved P and F (Skirrow et al., 2013; Jaireth et al., 2014). The above reactions may* **cause REEs to deposit in silicate minerals along the country rock interface (Anenburg & Mavrogenes 2018; Anenburg et al., 2020). eEnrichment of incompatible elements such as REEs, U, Th, Nb, Ba, Sr, Zr, Mn, Fe, Ti in the fluids occur due to liquid immiscibility, especially in liquids rich in alkalis which promote REE solubility** *(#10, 13, 14, 15, 16, 17, 18, 19 in Fig. 3 and Table 3; Cordeiro et al., 2010; Skirrow et at., 2013; Jaireth et al., 2014; Pirajno, 2015; Mitchell, 2015; Chakhmouradian et al., 2015; Stoppa et al., 2016; Poletti et al., 2016; Giovannini et al., 2017; Simandl and Paradis, 2018; Spandler et al., 2020;* **Anenburg et al., 2020**). *Carbonatite-alkaline complexes are often enriched in ferromagnesian minerals that cause well-defined magnetic and gravity anomalies (#9 in Fig. 3 and Table 3; Gunn and Dentith, 1997; Thomas et al., 2016). Fenitisation often enriches country rocks in K and Na (#12 in Fig. 3 and Table 3; Le Bas, 2008; Elliott et al., 2018).* **In alkali rich intrusions, LREEs are retained in the primary carbonatite while HREEs tend to concentrate in Fenites, particularly K-Fenites; whereas in silica-rich or alkali-poor intrusions, HREEs remain in the carbonatite (Anenburg et al., 2020). Size and HREE/LREE concentration of the fenites halo are a major proxy.***"

line 175: These late stage minerals are often in-situ replacement of two minerals: burbankite and carbocernaite, which are probably the two most important primary REE minerals in carbonatites, but because of their solubility, they are rarely preserved. Typically, currently observed REE minerals don't precipitate from hydrothermal fluids as you stage in your line 173, rather they are the result of local redistribution of REE after burbankite and carbocernaite are dissolved, and their REE component remain immobile. See for example the Anenburg et al 2020 paper I referred to earlier, and these two relevant papers by Kozlov et al (https://doi.org/10.3390/min10010073) and Andersen et al (https://doi.org/10.1016/j.oregeorev.2017.06.025)
Also please correct mineral name from "parasite" to "parisite".

**Response:** The added information and references have greatly improved the model and we are very grateful for your insightful inputs. We also thank you for pointing out the spelling error. It will be corrected in the next revision. The paragraph (lines 176 – 183) will also be modified as follows –

*"Rare earth element mineralisation in the carbonatites can be in the form of primary REE-bearing minerals (e.g., Mountain Pass, Mariano, 1989; Castor, 2008; Verplanck and Van Gosen, 2011; Van Gosen et al., 2017)* **or by secondary hydrothermal activity, including in-situ replacement, or from late magmatic fluid phases expelled from the carbonatite magmas** *(Verplanck and Van Gosen, 2011; Skirrow et at., 2013; Jaireth et al., 2014; Van Gosen et al., 2017). Primary REE-bearing cumulates include perovskite, pyrochlore, apatite and calcite, while late-stage REE-bearing minerals include bastnäsite,* **parisite, and synchysite that form from the redistribution of soluble primary phases such as ancylite, burbankite and carbocernaite** *(#24 in Table 3; Verplanck and Van Gosen, 2011; Skirrow et al., 2013; Van Gosen et al., 2017;* **Andersen et al., 2017; Anenburg et al., 2020; Kozlov et al., 2020**).*"

line 185: Something which should be highly relevant for your modelling is the decoupling between LREE and HREE in carbonatites and fenites. Our experimental study (Anenburg et al 2020 above) demonstrated that LREE are retained inside the carbonatite, whereas HREE tend to be mobilised outwards into fenites. This is also observed in nature: see example papers by Andersen et al and Broom-Fendley et al:
https://doi.org/10.2138/am-2016-5532
https://doi.org/10.1016/j.oregeorev.2016.10.019
https://doi.org/10.2138/am-2016-5502CCBY

**Response:** Thank you very much for your valuable suggestions. We have included it in the model and we think it has benefitted from your suggestion. We now have an

added vector towards mineralised zones which would be very relevant for camp scale exploration studies.

---

## Author Comment (AC2)

[revised manuscript text omitted]

---

## Author Response (AR1)

**Response to RC1**

We thank Reviewer 1 for the positive and constructive review and insightful comments and suggestions. We have addressed all comments and acknowledge that this has led to significant improvement of the manuscript.

1. This is a thorough and detailed study. The main strength of the study is the rigorous approach towards identifying all the relevant features from various datasets. However, such a data pre-processing workflow could also add artifacts or cause identification of same features multiple times. Given the lack of Figures representing input data or the final identified features, it is difficult to understand and relate geophysical signals to the extracted features.

   It is true that data-pre-processing workflows could result in the addition of artefacts. However, the use of a conjunctive operator (the fuzzy AND) in the first stage FIS for combining the various features extracted from the geophysical and topographic datasets ensured that only those features that were consistently present in all derivative maps were used as inputs for the prospectivity modelling. We think that this step had minimized artefacts as the features that are present in all different datasets are less likely to be artefacts. Further, as explained in section 6, pages 14-15, lines 266-273, the uncertainty likely to be introduced by pre-processing was taken into consideration while allocating confidence values.

2. The scale and resolution of the primary data is quite wide-ranging, therefore one question that arises is how these were integrated together while maintaining adequate balance between extracting information from the datasets but at the same time keeping a non-subjective and quantitative check on introduction of stochastic uncertainties. Data inconsistencies is a common issue in prospectivity mapping studies, but here there seems to be one-to-two-orders of differences in the spatial resolution of the input datasets, so the magnitude of artifacts could easily increase accordingly. Moreover, the spatial resolution of the predictors maps and prospectivity mapping is not provided in the manuscript.

   We agree that there are one-to-two-orders of difference in the spatial resolution of the datasets, particularly between gravity and magnetic datasets. However, as discussed in our response to Comment 1 above, we think that the use of features that are common across different datasets likely minimised artefacts caused by the scale variations.

   A grid-cell size of 3 km was used for both, predictor maps and prospectivity modelling. The grid-cell size was a trade-off between high-resolution

magnetic data and low-resolution gravity data. Also, it is approximately the average size of carbonatite-alkaline complexes in the study area. We thank the reviewer for pointing out this omission in the original manuscript. This has now been added in the revised manuscript (Section 5, page 10, lines 192-197).

3. Table 4 contains a lot of repetition and needs to be simplified. The full form of the acronyms used in Table 4 are not provided in the manuscript. Most are standard acronyms such as RTP (Reduced to Pole), but to conform to the norms of scientific writing, as a suggestion, it would be useful to define all the acronyms, either in the table as footnotes or in the text.

   We thank the Reviewer for the comment and suggestions. We agree that the original Table 4 was cluttered and contained redundant information because of repetitions. Table 4 (and other tables) have been revised and simplified, and the repetition of information has been minimized. The acronyms have been provided with their full forms. Some of the tables (Tables 1 and 6A, B and C of the original manuscript) have been moved to the appendix on the suggestion of Reviewers 2 and 3. This has improved the readability of the paper.

4. Several predictor maps are used more than once in the modelling procedure as shown in Table 5 and Table 6. Will this not increase the influence of such predictor maps in the results? The question is should they be really considered more than once, because spatially they are the same? From Section 5 it seems that the objective of using three FIS was to progressively reduce the area of exploration, but if large-scale features of a previous FIS are used in the next FIS, then how does this influence the results?

   Thank you for posing this question. Some of the predictor maps were used more than once in the modelling because these maps provided evidence for more than one targeting criteria. For example, the 'distance to rift' map is used to proxy for the extensional setting and the associated mantle plume in the FIS for delineating the prospective fertility/geodynamic settings. The same map is used as a proxy for fluid pathways in the second FIS for demarcating the favourable transport architecture. Wherever a predictor map is used for more than one component, different parameters are used for the fuzzy membership functions for each component. This has now been clarified in Tables 3 and 4.

   Moreover, the use of conjunctive operators (fuzzy AND and fuzzy PRODUCT) in both first and second stage FISs ensured that minimum of the influences of all inputs would be propagated to the output prospectivity map. This is also the reason why, unlike the weights-of-evidence or Naïve Bayesian

classifiers, the FIS approach is not significantly affected by conditional dependencies in the input datasets.

5. Overall, the research is well-implemented and concisely presented in the manuscript. Results are rationally evaluated and discussed.

   Thank you for the exhaustive review and the insightful comments.

**Response to RC2**

We thank Reviewer 2 for his positive assessment of the manuscript and constructive and insightful comments and suggestions. We have addressed all comments and suggestions in the revised manuscript.

In the manuscript with the title "REE's associated with carbonatite-alkaline complexes in western Rajasthan, India: exploration targeting at a regional scale", the authors present a mineral prospectivity approach to determine potential target sites for REE exploration. The work is based on extensive previous experience by the authors in the field of mineral prospectivity analysis and uses state-of-the-art methods, applied to a regional case study. As such, it is an interesting contribution to the field and suitable for a publication in a scientific journal, specifically for the special topic on the state of the art in mineral exploration.

Overall, the manuscript contains all elements that are relevant for a scientific publication, but I do have several concerns about the structure itself.

1. The manuscript contains an excessive amount of tables. Even if one can usually argue that tables help summarising information, it appears quite the opposite here: as a reader, one is constantly shifting between reading text in the main document, and reading text in the tables. In my point of view, this is really not helping and distracts from the main contribution of the manuscript. In addition, almost all tables contain a very high amount of redundant information (see comments below). I would strongly suggest to re-structure the document in a form that the relevant aspects are in the main text, and to place all tables (if they should be kept) in the appendix.

   We agree that the paper and tables needed restructuring. In the revised manuscript, we have reduced the number of tables within the body of the manuscript and moved some of them to an appendix. The revised manuscript has six tables. The original Tables 1 and 6A, B and C have been moved to the appendix. Also, all tables have been revised and simplified to improve the readability and to avoid cluttering and repetition of information.

2. Another point concerns the results section: this is almost non-existent (two short paragraphs) and actually a lot of outcomes are missing here. They are then described in the discussion (e.g., lines 318-334). Also here, I would

suggest to carefully revise results and discussion section to place the content at the part where a typical reader would expect them - this is to some extent always up to personal choice, but I think that a restructuring will definitely help clarifying the main contribution of the work.

We appreciate the suggestion and agree that the results and discussion sections in the original manuscript needed restructuring to improve the readability. In the revised version, the Results and Discussion have been combined in a single section, "Results and Discussion". The recommendations have been moved to the Conclusion section.

3. Concerning the scientific contribution: all main components of the used data sets are well described and justified. What is missing is a description of the approach itself (the fuzzy inference system). The authors refer to previous publications on the topic - but actually, they are partly paywalled and will not be accessible to all readers (I also could not find openly available preprints). I would suggest to include some more details into the manuscript. In the current form, it is very difficult to understand how exactly uncertainties are considered and which of the novel approaches are implemented. Also, the description would benefit from some of the base references (if I am not mistaken, then the approaches go back to the work of Duda in the 1970's).

Thank you for flagging this point. We agree that a complete description of the theory and implementation of fuzzy logic will undoubtedly add value to the paper. We completely understand the concerns regarding a background to the approach. To address this, we have cited some additional publications in the text in lines 61 – 63 on page 2 and also provided some examples of open-source applications that can be used to implement FIS in lines 199 – 206 on page 10. We have also cited the background available on the webpage of MathWorks® on FIS, which can be freely accessed at: https://in.mathworks.com/help/fuzzy/fuzzy-inference-process.html. We think that an exhaustive description of the FIS approach is not feasible and would shift the focus away from the paper's central idea, and a brief introduction cannot do justice to the approach.

4. In the interest of open science and in order to better understand the approach itself (and to make it reproducible), it would also be very beneficial to make the processing scripts available on an open repository (e.g, GitHub or an institutional repository).

In this study, we used the commercial software Fuzzy Logic Toolbox of MathWorks® to implement the model. The Monte Carlo simulations were implemented using a trial version of the proprietary software XLSTAT®.

However, we appreciate and agree with the comment of the Reviewer. We have provided some open open-source applications that can be used to implement FIS (Section 6, page 10, lines 199 – 206). These tools can be used to replicate the approach.

5. One key aspect of the approach is that the authors estimate uncertainties in the resulting prediction maps. This is an important extension, when compared to conventional prospectivity mapping approaches. It should be mentioned, thought, that the analysed systemic uncertainties only address a small part of the overall set of uncertainties (here called "systemic", similar to "epistemic"?): the uncertainties in the hyper-parameters of the distributions. Many other uncertainties which, I think, the authors would also group under the term "systemic" are not included (choice of the distribution, model structure, choice of using a FIS, …). To be sure, this is acceptable, as no approach yet exists to consider all uncertainties, but it should at least be mentioned that the uncertainty estimates are limited to this specific aspect.

We thank the Reviewer for pointing out this limitation. We have incorporated this comment on Page 12, Section 6, lines 237-239.

Further (minor) comments to specific sections in text (identified by line numbers):

75: In the UQ literature, the terms epistemic and aleatory are commonly used. How do the terms systemic and stochastic refer to these?

As clarified on Page 3, Section 1, lines 64-70, the term systemic corresponds to epistemic and stochastic corresponds to aleatory.

97: Please describe the geologic/ geodynamic setting instead of a list of continents and countries.

Thank you for the suggestion. We have added the geodynamic setting in lines 88-89, page 3, section 1.

147: As the presented work strongly depends on the mineral system model, it would be important to include the full reference here - or, at least, to provide an accessible version of a preprint (if the paper is not yet accepted). In the current form, this model cannot be evaluated.

We agree. However, as a preprint of the paper containing the exhaustive details of the mineral systems model is currently unavailable, the model has been summarised briefly in section 4. We have provided all the relevant references.

Table 4B, C: highly redundant information - see general comments above.

As suggested, the table has been revised. Please see Table 3 in the revised manuscript.

Table 5: last column: why Piece-wise linear twice? Overall, also highly redundant.

As suggested, the table has been revised. Please see Table 4 in the revised manuscript.

Fig. 4: Also here, a lot of redundant information. A more compact description would help for a clearer representation.

We have modified Figure 4 in the revised version.

233, FIS-model(s): Were the three separate models chosen on basis of a geodynamic/ mineral systems consideration or on basis of the operator functions (into "AND" and "Product" branches)?

The three separate FIS models are used to combine the respective predictor maps of the three essential components of the mineral systems model (namely, Fertility/geodynamic setting, transportation and emplacement architecture) in order to generate fuzzy prospectivity maps for the respective component. We used the conjunctive fuzzy AND operator in these three FIS to combine the respective predictor maps.

The second stage FIS was used to combine the above three sets of fuzzy prospectivity maps for fertility/geodynamic settings, transportation and emplacement architecture using the fuzzy product operator to generate the final prospectivity map for REEs.

The above has been clearly explained in Section 6, page 14, lines 252-262.

Fig. 5: scale, colorbar, missing, difficult to interpret. Suggestion: include subfigures for detailed areas and include description in results section (instead of discussion).

Thank you for pointing it out. We have included these critical elements in the revised figures.

**Response to RC3**

We thank the Reviewer for the positive comments and welcome the constructive comments. We have addressed all his comments and suggestions in the revised version. Additionally, we have provided our responses in the attached review.

The paper REEs associated with carbonatite-alkaline complexes in western Rajasthan, India: exploration targeting at regional-scale by Malcolm Aranha, Alok Porwal, Manikandan Sundaralingam, Ignacio González-Álvarez, Amber Markan and Karunakar Rao3 provides a study that will interest not only researchers into mineral resources spatial data modelling but also researchers into REE mineralisation and industry explorationists and it appropriate for the Journal.

The paper describes the study and results in a well written and structured paper. There are minor edits that are provided in the attached review of the paper.

1. The main issue with the paper is the number and length of tables that make reading the paper difficult. The information in the tables, however, is very useful and should be included. I would suggest that some of the larger tables are moved to either appendices or a supplementary data section. This will improve the structure and readability of the paper without losing important information.

   We agree that the tables in the original manuscript were too cluttered and repetitive and needed restructuring. In the revised version, we have shortened, simplified and revised Tables 3 and 4 (Tables 4 and 5 in the original manuscript). The original Tables 1 and 6A, B and C have now been moved to the appendix.

2. There are some improvements that can be made to the description of the targets and the Discussion and Recommendations section which are highlighted in the attached review.

   We have addressed the comments and suggestions in the revised version and responded to the individual comments in the attached review.

3. There are minor issues with the references in the paper with some cited and not referenced and vice versa. Please see the attached.

   Thanks for pointing them out. We have addressed all the references related issues in the revised manuscript. The cited references on page 5 and Fig. 3 that were previously missing from the references list have now been included.

4. If these issues are addressed the paper can be published.
   We thank the Reviewer for his positive review and comments.

---

## Author Response (AR2)

We thank the editor for handling our manuscript and for recommending our manuscript for publication. We are also grateful for the constructive comments and suggestions. We have addressed each of them below.

congratulations to the manuscript!
I found only very minor things to correct:
- l 122: Réunion is a name of a place. I think, even the plume is written with as own name (starting with Cap) and with the é.

Thank you for pointing this out. We agree with the suggestion and made the necessary changes in line 122 to rectify the error.

- l 136: Is this manuscript still in review? Please update, if something changed meanwhile.

Yes, this manuscript is still under review.

- l 160: One of the few sentences with complicated structure. Please make it easier to read

We have now rephrased the sentence in line 160 to make it simple.

- section 6.1.: for readability: please reduce the number of phrases inside rounded brackets

Thank you for the suggestion. We have reduced the number of phrases inside rounded brackets in section 6.1.

---

## Author Response (AR3)

**Remarks from the preceding review file validation**

Please note that coloured table cells are not allowed in the final paper. If possible, please rotate tables and figures into portrait mode.

**Response:** In accordance with the comments of the technical check team, colours have been removed from the tables, and all tables and figures in the main paper have been changed to portrait mode. As a result, the placement and formatting of some figures and tables have changed. However, the content of the paper remains exactly the same as accepted by the topical editor.